**EMBO** *reports*

# GPNMB marks a quiescent cell population in melanoma and promotes metastasis formation

Fiorenza Lotti [1], Marine Melixetian [1], Thalia Vlachou [1,5], Marco S Nobile [2], Leone Bacciu[2], Marco Malferrari [3], Nicolò Quaresima [3], Stefania Rapino[3], Federica Marocchi [1], Massimo Barberis [1], Chiara Soriani [1], Barbara Gallo[1], Velia Mollo[1], Ilaria Ferrarotto[1], Daniela Bossi[1,6], Pier Francesco Ferrucci [1,7], Pier Giuseppe Pelicci [1,4], Lucilla Luzi [1] & Luisa Lanfrancone [1]✉

## Abstract

**Melanoma exhibits high intratumoral heterogeneity, characterized by a diverse population of cells undergoing dynamic transitions between cellular states. These adaptive changes enable melanoma cells to survive in the harsh tumor microenvironment, acquire drug resistance, and metastasize. One such state, quiescence, has been linked to both relapse and drug resistance, but its underlying biology and molecular mechanisms remain poorly understood. Our study challenges the conventional understanding of melanoma quiescence. Contrary to the notion of a rare, unique subpopulation, we demonstrate that quiescence is a highly dynamic state accessible to most, if not all, melanoma cells. This state is exquisitely sensitive to microenvironmental cues. We identify GPNMB as a marker of quiescence, that is expressed in both primary and metastatic tumors. GPNMB-positive cells exhibit a pro-metastatic phenotype and are enriched in metastatic sites, suggesting a potential role for quiescence in tumor dissemination. Our findings position GPNMB as a valuable marker for isolating quiescent melanoma cells and as a potential therapeutic target to tackle metastasis.**

**Keywords** Melanoma; Quiescence; Metastasis; Hypoxia; Target Therapy
**Subject Categories** Biomarkers; Cancer; Cell Cycle

## Introduction

Metastatic spreading represents the most frequent cause of death for melanoma patients and a major challenge for oncologists (Schadendorf et al, 2018). Melanoma heterogeneity and plasticity facilitate metastatic dissemination. Melanoma cells exhibit significant phenotypic and functional diversity, enabling them to adapt to the tumor microenvironment (TME) and acquire novel mechanisms of survival, dissemination and drug resistance upon treatment (Diazzi et al, 2023; Tirosh et al, 2016; Hossain and Eccles, 2023; Rambow et al, 2019; Arozarena and Wellbrock, 2019; Huang et al, 2021a; Quintana et al, 2010). Despite the wide range of accumulating mutations that lead to the emergence of diverse subclonal populations in melanoma, the response to environmental pressure can influence the expression of genes involved in cell proliferation, immune evasion and metastasis (Grzywa et al, 2017; Motwani and Eccles, 2021). These epigenetic changes can vary in different melanoma cells within the same tumor, further contributing to the ability of melanoma cells to dynamically change their phenotype (Rambow et al, 2019).

This transition from one phenotype to another, primarily driven by interactions and secretory activities between cells and TME, has been described in preclinical models and clinical samples (Tirosh et al, 2016; Hoek et al, 2008; Hoek and Goding, 2010). Recent studies have identified distinct functional cell states within melanoma, characterized by unique transcriptional profiles and spatial organization within the tumor (Hoek et al, 2008; Hoek and Goding, 2010; Karras et al, 2022; Rambow et al, 2018; Davis et al, 2019). These state changes enable melanoma cells to reorganize hierarchically, contributing to varying degrees of primary tumor growth and metastatic spread (Rambow et al, 2019; Karras et al, 2022; Rambow et al, 2018; Wouters et al, 2020; Tsoi et al, 2018). Furthermore, this dynamic tumor environment can give rise to various drug-tolerant states, impacting therapeutic sensitivity and posing significant challenges for long-term management, ultimately leading to clinical failure (Rambow et al, 2018). Despite extensive research leading to the identification of various functional states in melanoma cells, biological properties and molecular mechanisms of different cell subpopulations in melanomas remains largely unexplored.

Quiescent cells have been described in different tumor types as a relatively small-sized subpopulation within the tumor characterized by a temporary and reversible cell cycle arrest (Nabil et al, 2021). Unlike senescent cells, which are in an irreversible cell cycle-restricted state,

[1]Department of Experimental Oncology, European Institute of Oncology IRCCS, Milan, Italy. [2]Department of Environmental Science, Computer Science and Statistics, University of Ca' Foscari, Venice, Italy. [3]Department of Chemistry "Giacomo Ciamician", Università di Bologna, Via Selmi 2, 40126 Bologna, Italy. [4]Department of Oncology and Hemato-Oncology, Università degli Studi di Milano, Milan, Italy. [5]Present address: Purposeful, Tritis Septembriou 144, Athens 11251, Greece. [6]Present address: Institute of Oncology Research, Oncology Institute of Southern Switzerland, Bellinzona, Switzerland. [7]Present address: Department of Oncology, Gruppo MultiMedica, Milano, Italy.
✉E-mail: luisa.lanfrancone@ieo.it

quiescent cells can re-enter the cell cycle and are thought to play a crucial role in driving tumor relapse and drug resistance in many cancers (van Velthoven and Rando, 2019).

In melanomas, quiescent cells have been shown to exhibit characteristics similar to slow-cycling cells, including reduced proliferation, increased invasiveness and metastatic potential, resistance to treatment (Roesch et al, 2010, 2013; Perego et al, 2018; Puig et al, 2018; Moore et al, 2012; La et al, 2021). Slow-cycling cells have been shown to exhibit a reversible G2/M cell cycle arrest and low expression of KI67 (Roesch et al, 2010, 2013; Perego et al, 2018; Puig et al, 2018; Moore et al, 2012; La et al, 2021) unlike quiescent cells that are typically arrested in a specific G0 state and do not express the KI67 marker (van Velthoven and Rando, 2019). Slow-cycling cells were also reported to possess the unique ability to dynamically reprogram themselves in response to cues from the surrounding microenvironment (Hoek and Goding, 2010; Fattore et al, 2020; Ahn et al, 2017; Shiokawa et al, 2021; Basu et al, 2022; Antonica et al, 2022; Razi et al, 2023).

These cells have been identified using various markers, including CD133 (Simbulan-Rosenthal et al, 2019), aldehyde dehydrogenase 1 (ALDH1) (Wei et al, 2022), CD271 (Civenni et al, 2011; Boiko et al, 2010), ABCB5 (Quintana et al, 2010; Luo et al, 2012; Schatton et al, 2008), CD44 (Thapa and Wilson, 2016; Wang et al, 2018), lysine demethylase 5B (JARID1B) (Roesch et al, 2010) and SOX10 (Capparelli et al, 2022). However, the reliability of these markers for identifying a truly quiescent cell population remains questionable (Parmiani, 2016; Prasmickaite et al, 2010). Alternatively, slow-cycling cells have been identified using labeling-retaining assays, which track the retention of molecules like H2B-GFP or PKH26 during proliferation (Puig et al, 2018; Perego et al, 2018; Roesch et al, 2010). Nonetheless, these assays are limited by their inability to determine the hierarchical relationship between quiescent cells and other tumor subpopulations, as they rely on the proliferative history of tumor cells rather than their dynamic relations. Together, these limitations have hindered the characterization of quiescent cells, which is crucial for identifying their specific dependencies and developing targeted therapies.

In this study, we performed an extensive characterization of quiescent cells in primary human melanomas and metastases, combining label-retaining and surface marker approaches for their isolation. Our data demonstrate that rather than being a small-sized subpopulation, quiescence in melanoma is a functional state shared by most, if not all, melanoma cells. We identified a strategy to specifically target this state, leading to a reduction in the pro-metastatic potential and development of drug resistance in melanoma.

# Results

## Label retention and cell cycle dynamics identify quiescence as a dynamic trait of melanoma cells

To visualize non-dividing melanoma cells in vivo, we leveraged a doxycycline-repressible histone H2B-GFP reporter system integrated into the DNA of melanoma cells (Appendix Fig. S1A). Doxycycline treatment suppresses H2B-GFP expression, allowing serial dilution of the GFP signal in proliferating cells and its retention into non-dividing cells (Puig et al, 2018; Miller et al, 2018;

Falkowska-Hansen et al, 2010). Melanoma cells derived from two PDXs harboring NRAS (MM13) or BRAF (MM27) mutations were engineered to express H2B-GFP via lentiviral transduction. Cells were then serially transplanted into NSG mice three times. Following each transplant, the GFP-positive (GFP +) population was isolated using fluorescence sorting to obtain a homogenous population expressing H2B-GFP (Appendix Fig. S1B). These GFP+ cells were then transplanted into recipient NSG mice and allowed to grow into palpable tumors (MM13, Fig. 1A; MM27, Appendix Fig. S1C). Doxycycline was administered for 3 additional weeks, resulting in a gradual decrease of the GFP+ cell population up to 0.2–1% after 3 weeks (Fig. 1A; Appendix Fig. S1C,D). ProCell framework, a model-driven approach based on a stochastic simulation of cell proliferation, allowed us to investigate population dynamics of GFP+ cells (Nobile et al, 2019; Nobile et al, 2019). Providing flow cytometry data of the initial and the target GFP-fluorescence, ProCell tested 4 different cell proliferation scenarios (described in "Methods") possibly representing the melanoma intra-tumor heterogeneity. The parameters of each model were calibrated against experimental fluorescence histograms obtained after 1 week of doxycycline chasing. Models 2, 3 and 4 achieved a better fitting to the experimental data compared to model 1 (that is based on the assumption of a single cell population with varying cell cycle properties), which was deemed insufficient to explain the melanoma intricate proliferation dynamics (Appendix Fig. S2A). We further validated our models using best parameterization and simulating the various configurations 100 times each, at week 1 (168 h)—the same window as for model calibration—and at week 3 (504 h) (Appendix Fig. S2B). This approach allowed us to account for the randomness introduced by the stochastic initialization of cell populations. Model 4 (based on the assumption of two populations, slow- or fast-proliferating respectively) failed to accurately reproduce the experimental data at week 3, while models 2 and 3 provided solutions of similar quality (Fig. 1B). Both models assume the presence of a quiescent population (with either one or two proliferating populations), albeit model 2 introduces less complexity and better fits in the short (1 week) period (Fig. 1B). Thus, the model that best fits and represents melanoma proliferation kinetics in our model system is model 2, that accounts for the presence of one population of quiescent cells (defined as cells that remain undivided for the 504 h of chasing; at an estimated proportion of 31%) and one of proliferating cells (estimated at 44.94 ± 19.71 h, in a range between 21 and 504 h; estimated proportion of 69%).

To confirm the presence of G0 quiescent cells within the label-retaining pool, we assessed the proliferative potential of the GFP+ population by cell cycle analysis. KI67 expression was used as a marker of active G1-S-G2M cell cycle phases, while BrdU incorporation was used as marker of DNA synthesis (24 h pulse). The majority of GFP+ cells were negative for both KI67 (Fig. 1C; Appendix Fig. S1E) and BrdU incorporation (Fig. 1D; Appendix Fig. S1F), indicating their quiescent (G0) state. However, a distinct subpopulation (~40%) expressed KI67 (Fig. 1C; Appendix Fig. S1E), revealing ongoing cell cycle activity within the GFP+ pool, with some cells undergoing DNA synthesis (~20% BrdU +) (Fig. 1D; Appendix Fig. S1F). Since KI67 levels progressively increase during the G0-G1 transition (Miller et al, 2018; Alessio et al, 2021), the presence of KI67+ cells suggest the existence of a fraction of label-retaining cells re-entering the cell cycle, with some BrdU+ cells

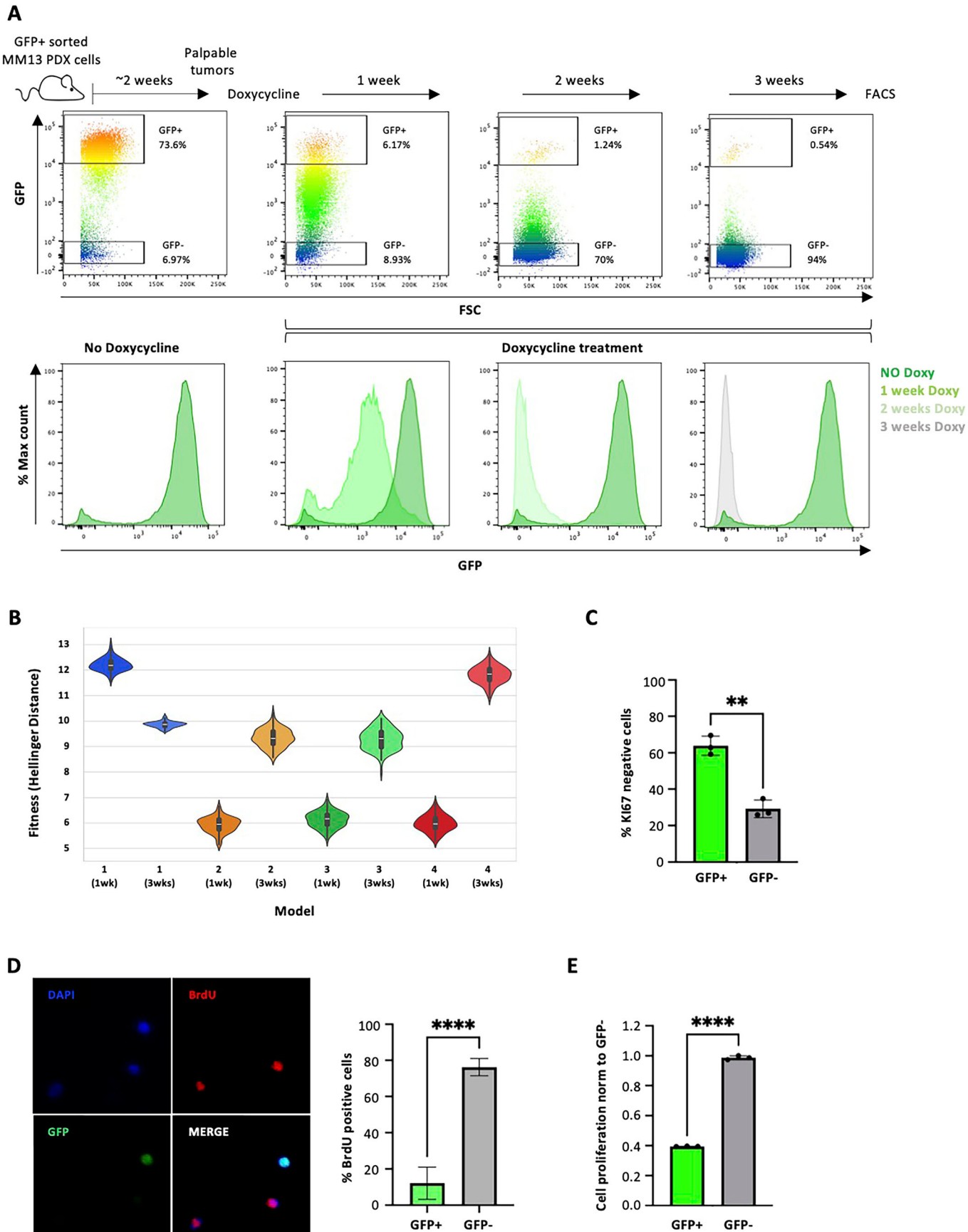

**Figure 1. Isolation of label-retaining MM13 PDX cells in vivo by H2B-GFP tet-off vector expression.**

(A) H2B-GFP MM13 cells induced in vivo with doxycycline. GFP-expressing, FACS-sorted MM13 PDX cells ($2 \times 10^5$/mouse) injected in NSG mice grown for 2 weeks and then treated for 3 weeks with doxycycline. Representative dot plot flow cytometry analysis (upper panel) and distribution histograms of GFP expressing cells (lower panel) showing cell proliferation (gradual GFP loss) over time with around 0.5% GFP+ cells remaining after 3 weeks of treatment. Not treated cells represent the positive control (no doxycycline). (B) Violin plot of the fitness values (Hellinger distance) obtained using best parameterization and simulating the various configurations of the 4 in silico proliferation models 100 times each, at week 1 (168 h) and at week 3 (504 h). 1 (1 wk): min 11.578267, max 13.058745, median 12.209594, 25%: 12.029869, 50%: 12.186589, 75%: 12.372117; 1 (3 wks): min 9.596297, max 10.201864, median 9.862849, 25%: 9.786830, 50%: 9.869009, 75%: 9.927403; 2 (1 wk): min 5.149632, max 6.660758, median 5.928789, 25%: 5.731673, 50%: 5.949385, 75%: 6.154683; 2 (3 wks): min 8.566041, max 10.328199, median 9.339799, 25%: 9.092175, 50%: 9.321489, 75%: 9.607174; 3 (1 wk): min 5.335960, max 6.926181, median 6.125735; 25%: 5.921229, 50%: 6.165732, 75%: 6.335626; 3 (3 wks): min 7.987454, max 10.101414, median 9.281341; 25%: 8.958406, 50%: 9.313066, 75%: 9.591875; 4 (1 wk): min 5.291155, max 6.814217, median 5.984589; 25%: 5.784939, 50%: 5.964859, 75%: 6.202167; 4 (3 wks): min 10.904812, max 12.630359, median 11.811980; 25%: 11.593972, 50%: 11.845900, 75%: 12.062802. (C) GFP+ and GFP- MM13 cells, after 3 weeks of doxycycline treatment, were stained with KI67 and analyzed by flow cytometry. Data are presented as mean ± SD ($n = 3$ biological replicates). Student $t$ test was used (**$P = 0.00109$). (D) GFP+ and GFP- MM13 cells, after 3 weeks of doxycycline treatment, were pulsed with BrdU for 24 h in vitro. Representative images of cells fixed and analyzed by confocal microscopy and counted for GFP and BrdU expression by ImageJ. Data are presented as mean ± SD ($n = 5$ technical replicates). Student $t$ test was applied to assess the significance (****$P = 3.30e$-10). Scale bar = 200 μm. (E) GFP + MM13 cell proliferation, after 3 weeks of doxycycline treatment, was assessed by CyQuant and normalized to GFP− cells, as mean ± SD ($n = 3$ technical replicates). $P$ values are based on unpaired Student's $t$ test (****$P = 1.85e$-07). Source data are available online for this figure.

already in S phase. Indeed, the GFP+ population disappears if treatment with doxycycline is prolonged beyond 3 weeks (Appendix Fig. S1G). These findings strongly suggests that GFP+ cells have a high capacity to re-enter the cell cycle, and that most, if not all, GFP+ quiescent cells retain the ability to resume proliferation.

Cell cycle analyses of the GFP-negative (GFP-) population (Fig. 1C,D; Appendix Fig. S1E,F) revealed a predominantly KI67+ and short-pulsed BrdU+ phenotype, indicating a hyper-proliferative state. Consistent with this observation, CyQuant analysis confirmed a markedly higher proliferation rate in the GFP- population compared to the GFP+ population (Fig. 1E; Appendix Fig. S1H). Interestingly, a significant fraction of cells remained cell cycle restricted (~30% KI67− and ~10% short-pulsed BrdU−), suggesting their ability to re-enter quiescence despite their overall high proliferation rate.

Together, our findings confirm the existence of a small subpopulation of label-retaining cells within human melanomas, as previously reported (Roesch et al, 2010; Roesch et al, 2013; Perego et al, 2018; Puig et al, 2018). However, we demonstrate that this population comprises genuine quiescent cells capable of re-entering the cell cycle, and that quiescent cells are also present within the label-negative population, challenging the notion that label retention marks a distinct subpopulation of slow-cycling cells (Roesch et al, 2010; Roesch et al, 2013; Perego et al, 2018; Puig et al, 2018). Instead, our results suggest that label retention signifies a functional state that is potentially shared by most, if not all, melanoma cells. Notably, mathematical modeling of label-retaining cell dynamics, combined with KI67/BrdU analyses, indicates that approximately 30% of melanoma cells reside in a quiescent state under steady-state conditions.

## Quiescent cells from label-retaining or label-negative cells exhibit similar transcriptional profiles, including upregulation of genes involved in the G0 state and hypoxia response

We used scRNA-seq to analyze the transcriptional phenotype of quiescent cells from GFP+ and GFP- populations. These cells were purified from three primary tumors (3 weeks doxycycline treatment) and four matched lymph node metastases (6 weeks doxycycline treatment), whose functional characterization is subsequently detailed in Fig. 7. First, we analysed the whole GFP + or GFP− populations. Unsupervised Louvain clustering analyses and UMAP (uniform manifold approximation and projection) dimensionality reduction identified 9 transcriptional cell clusters (Fig. 2A). Although enrichment of GFP+ cells was observed in cluster 0 and 3 (Fig. 2B, right panel), there was minimal segregation between GFP+ and GFP− cells across the identified clusters (Fig. 2B, left panel), suggesting limited overall transcriptomic differences between these populations. Gene set enrichment analysis (GSEA) of GFP+ vs GFP- differential expressed genes (DEGs) revealed significant enrichment of genes involved in the regulation of cell death, cellular response to stress, DNA damage checkpoint, and cell cycle control (Fig. 2C). Analyses of several published melanoma signatures showed increased expression of invasion-related pathways ("MITF targets", "Invasion" and "MSC" melanoma signatures) in GFP+ cells, and of proliferation- and melanocyte-associated pathways ("Mitotic" and "Pigmentation" signatures) in GFP− cells (Appendix Fig. S3A).

To identify the quiescent compartment within the scRNAseq dataset, we annotated melanoma cells according to their cell cycle phase, using the Seurat package, and levels of KI67 expression. Cells showed a clear separation on UMAP projections based on cell cycle phase and KI67 expression score (Fig. 2D). Cells segregated by the median of KI67 expression showed a clear cell cycle association (Fig. 2E): KI67− cells were mainly found in the G1 phase, while KI67+ cells in the S and G2/M phases (Fig. 2D,E). Consistently, clusters 0, 1 and 8 were primarily constituted by G1 cells with low KI67 expression, whereas the remaining clusters contained S phase (clusters 2, 6–7) or G2/M phase (clusters 3–5) cells with variable levels of KI67 (Appendix Fig. S3B). GSEA analyses of cluster marker genes (Fig. 2F) showed enrichment of pro-invasive pathways in the G1/KI67−-related clusters (0, 1, 8), including iron ion homeostasis (Jung et al, 2019; Brown et al, 2020; Torti and Torti, 2013), cell response to oxygen levels (Roesch et al, 2010; Fattore et al, 2020; O'Connell et al, 2013; Huang et al, 2021a; Roesch et al, 2013; Raimondi et al, 2020) and mitochondrial electron transport (Huang et al, 2021a; Roesch et al, 2013; Raimondi et al, 2020). Conversely, S/G2M-KI67+ related clusters (2–7) revealed enrichment of pathways involved in the regulation of cell cycle progression (Fig. 2F). Interestingly, cluster identity and cell cycle phase assignment showed poor correlation with GFP

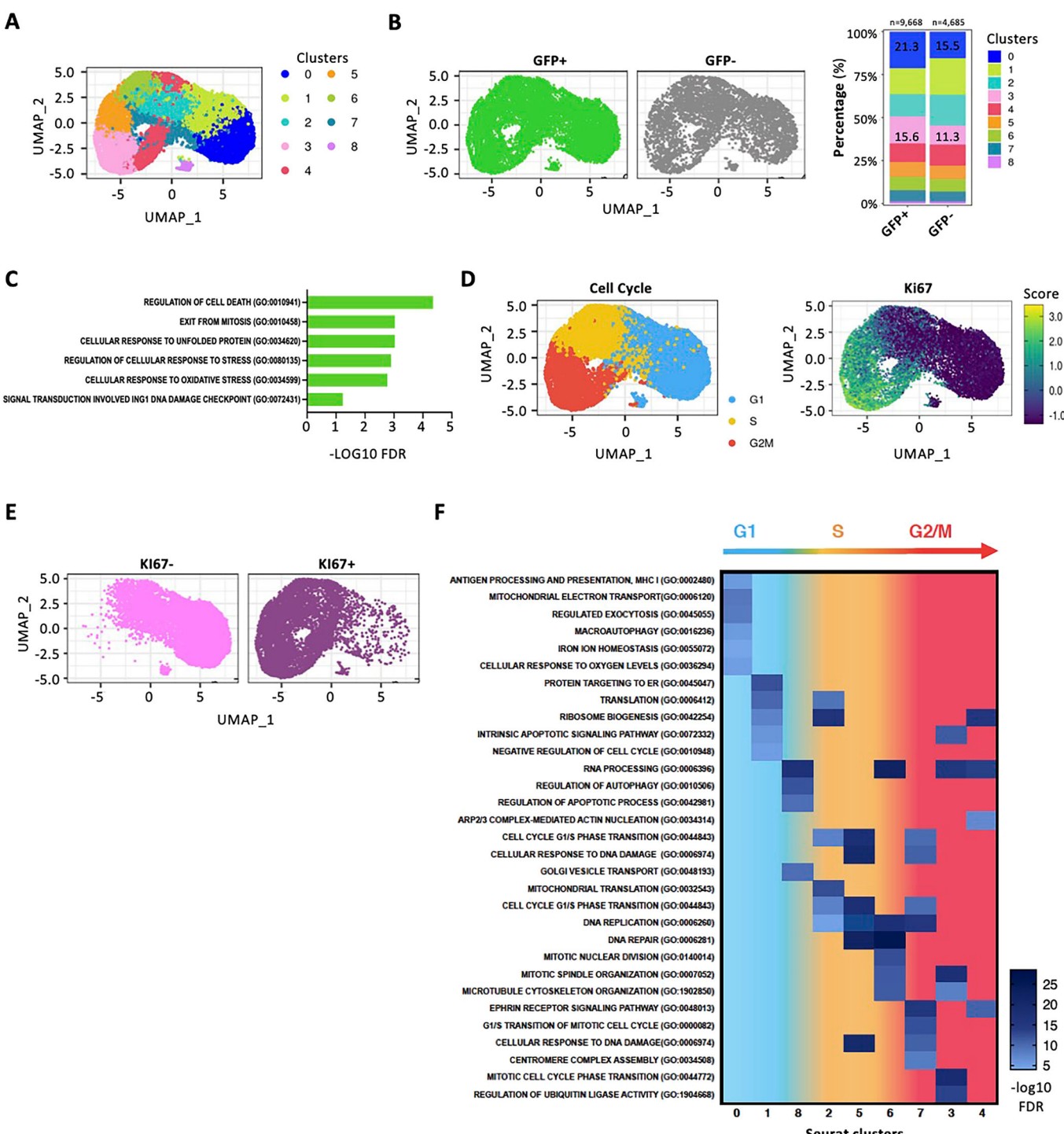

**Figure 2. Single-cell transcriptional landscape of primary and metastatic MM13 melanoma.**

(A) UMAP visualization of 14,353 MM13 melanoma cells analysed by scRNA-seq and integrated across 3 different melanoma primary tumors with 4 matched lymph node metastatic lesions: cells are colored by Seurat clusters. (B) UMAP projection of GFP+ and GFP− compartments of MM13 melanoma populations (left) and stack plot (right) of relative proportions of GFP+ and GFP− cells across Seurat clusters. (C) Enrichment pathway analysis (GO Biological Processes) of the GFP+ vs GFP− population. (D) UMAP visualization of MM13 melanoma cells colored by Seurat cell cycle annotation (left) and KI67 expression score (right). (E) UMAP visualization of MM13 melanoma cells grouped on median of KI67 expression. (F) Enrichment pathway analysis (GO) of the Seurat clusters in the GFP+ population. Pathways enriched at False Discovery Rate (FDR) less than 0.25 are shown.

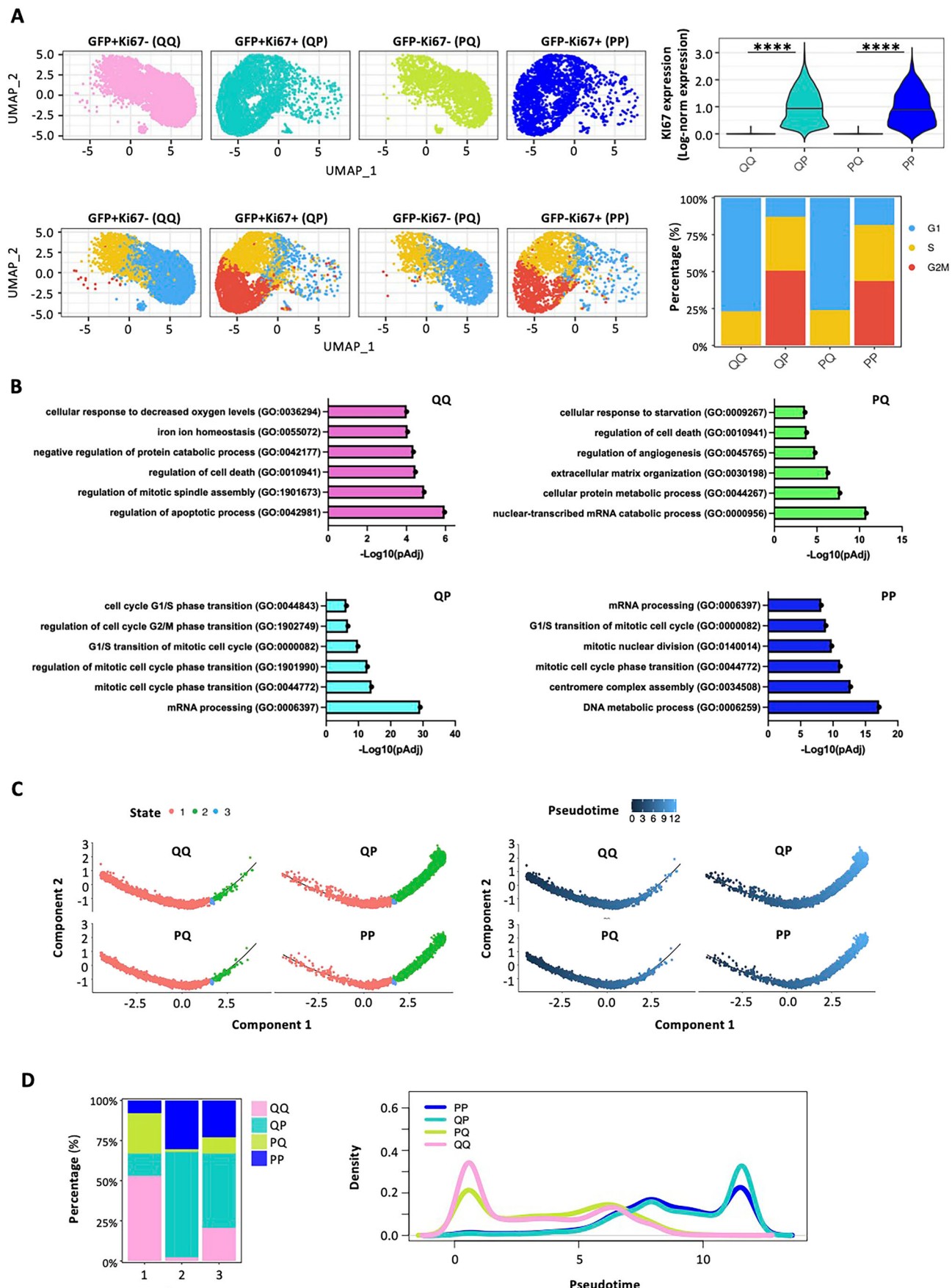

**Figure 3.   Single cell transcriptomic characterization of four cell populations grouped according to GFP fluorescence and KI67 expression.**

(A) Upper panel: UMAP visualization of the four states grouped according to their quiescent/proliferative profile: GFP + KI67− (QQ: 33.8%), GFP-KI67− (PQ: 33.6%), GFP + KI67+ (QP: 16.2%), GFP-KI67+ (PP: 16.4%), and violin plot of the KI67 expression (upper panel; Mann–Whitney non-parametric test was applied to assess the significance ($n = 14358$; ****$P = 2.2e-16$). Lower panel: UMAP visualization colored by Seurat cell cycle annotation with stack plot. Percentage of G0-G1 cells in QQ, QP, PQ and PP states were 77, 13, 76 and 18%, respectively. (B) Enrichment pathway analysis (GO) of the populations defined in (A). Kolmogorov–Smirnov test was applied. (C) Trajectory analysis of QQ, QP, PQ and PP states based on the expression of the top 20 variable genes. The trajectories are colored by state (left panel) and pseudotime (right panel). State 1 was selected as root state. (D) Stack plots (left) showing relative proportions of the four populations across pseudotime states; density plots (right) showing distribution of the four populations along pseudotime.

expression. Clusters 0 or 3 were enriched in GFP+ cells but displayed heterogenous cell cycle distribution, with cluster 0 containing mainly G1/KI67− and cluster 3 composed primarily of G2-M/KI67+ cells (Fig. 2B; Appendix Fig. S3B). This observation further underscores the heterogeneous composition of the GFP + population.

Finally, to characterize the transcriptional states of quiescent (G0) cells within the GFP+ or GFP- populations, we used KI67 expression to identify (Fig. 3A) GFP + KI67− (QQ) or GFP-KI67− (PQ) cells, enriched in G0/G1 phase and representing the quiescent cells of the GFP+ or GFP− populations, respectively; and GFP + KI67+ (QP) or GFP-KI67+ cells (PP), primarily in S/G2M phase and representing proliferating cells of the GFP+ or GFP−, respectively. Notably, the quiescent QQ and PQ cells exhibited high transcriptional similarity (Fig. 3A). Marker genes for each state (Dataset EV1) were then used to interrogate databases of known gene signatures/pathways. Confirming our previous observations, the quiescent QQ and PQ states exhibited great similarities, setting them apart from the other two states, which themselves shared some common features. QQ and PQ cells were characterized by enrichment of genes involved in regulating cell death and energy-depletion responses (hypoxia and starvation responses, respectively) (Fig. 3B) and increased expression of growth inhibitory genes (MODULE_488, obtained from MSigDB) (Appendix Fig. S4A). In contrast, the proliferating QP and PP states showed enrichment of genes associated with cell cycle regulation and reduced expression of growth inhibitory genes (Fig. 3B; Appendix Fig. S4A). Analyses of curated signatures showed significantly higher levels of expression of genes associated with the QQ signature (Appendix Fig. S4B; Dataset EV1), the integrated stress response (known to promote quiescence), and cell cycle inhibitors (Appendix Fig. S4C) in the QQ and PQ cells, as compared to QP and PP cells.

Finally, we investigated whether quiescence is linked to therapy resistance and relapse in melanoma patients, as expected for a population which has high adaptive properties and fuels the compartment of highly proliferating melanoma cells. Thus, we analysed the predictive power of the QQ signature to therapy response, taking advantage of an RNA-seq dataset of melanoma patients pre- and post-RAF/MEKi treatment (Data Ref. GEO dataset GSE77940 (Wagle et al, 2014) and two scRNA-seq datasets of patients before and after immune checkpoint inhibitors treatment (Data Ref. GEO dataset GSE115978). Notably, GSEA analysis showed enrichment of QQ signature in relapsed patients treated with MAPK and immune checkpoint inhibitors (Appendix Fig. S4D). Moreover, analyses of a recently published melanoma signature from patient biopsies (Pozniak et al, 2024) showed that our QQ/PQ cell states are enriched in the expression of melanoma

states associated to acquired resistance to immunotherapy (Appendix Fig. S4E). These data suggest that the QQ signature is predictive of poor therapy response and might contain valuable targets for the development of treatments against therapy-resistant melanoma cell populations.

Taken together, scRNAseq analyses of different subpopulations demonstrated that GFP+ and GFP− cells are heterogeneous cell populations, both containing quiescent (KI67-negative) and proliferating (KI67-positive) cells, albeit in different proportions. Moreover, quiescent cells from GFP+ and GFP− populations shared similar transcriptional properties, characterized by the expression of quiescence genes, stress-response genes (hypoxia and energy deprivation), and cell cycle inhibitor genes.

To gain insights into the hierarchical relationships across the four cell states, we performed a pseudotemporal trajectory analysis. This analysis identified three distinct pseudotime states. State 1, consisting of genes expressed in the early phase of the trajectory, was enriched in QQ and, to a lesser extent, in PQ cells (Fig. 3C). Conversely, State 2 was enriched in QP and PP cells. Similar results were obtained by analyzing cell distribution across pseudotime, with QQ/PQ and PP/QP cells showing similar density distribution along the pseudotime axis (Fig. 3D). Collectively, these data suggest that quiescence and proliferation are dynamic states in melanoma cells, with cells that are able to exit and/or re-enter the cycle, transiting from a quiescent (QQ) to an active proliferative state (PP and QP), and back to a quiescent (PQ) state.

In conclusion, by integrating KI67 expression/cell cycle analysis and scRNA-seq, we identified distinct cell states with unique transcriptional profiles and functional properties. Notably, the QQ and PQ states displayed hallmarks of true quiescence, characterized by enriched expression of stress response pathways and cell cycle inhibitors. Pseudotemporal trajectory analysis further supported the dynamic nature of these states, suggesting a potential transition from quiescence to proliferation within the GFP+ and GFP− populations.

## The quiescence state supports the metastatic phenotype

Previous studies in various tumor cell lines, including melanoma, have shown that label-retaining cells exhibit enhanced migration, invasion, and treatment resistance, as well as dependence on mitochondrial oxidative phosphorylation (La et al, 2021; Fattore et al, 2020; Basu et al, 2022; Kumar et al, 2021; Avagliano et al, 2020; Davis et al, 2019). To extend these findings to patient-derived melanoma cells, we sorted GFP+ and GFP− cells from MM13 and MM27 PDX. As compared to GFP- cells, GFP+ cells exhibited increased migration (Fig. 4A; Appendix Fig. S5A) and invasion potential (Fig. 4B; Appendix Fig. S5B), and decreased sensitivity to

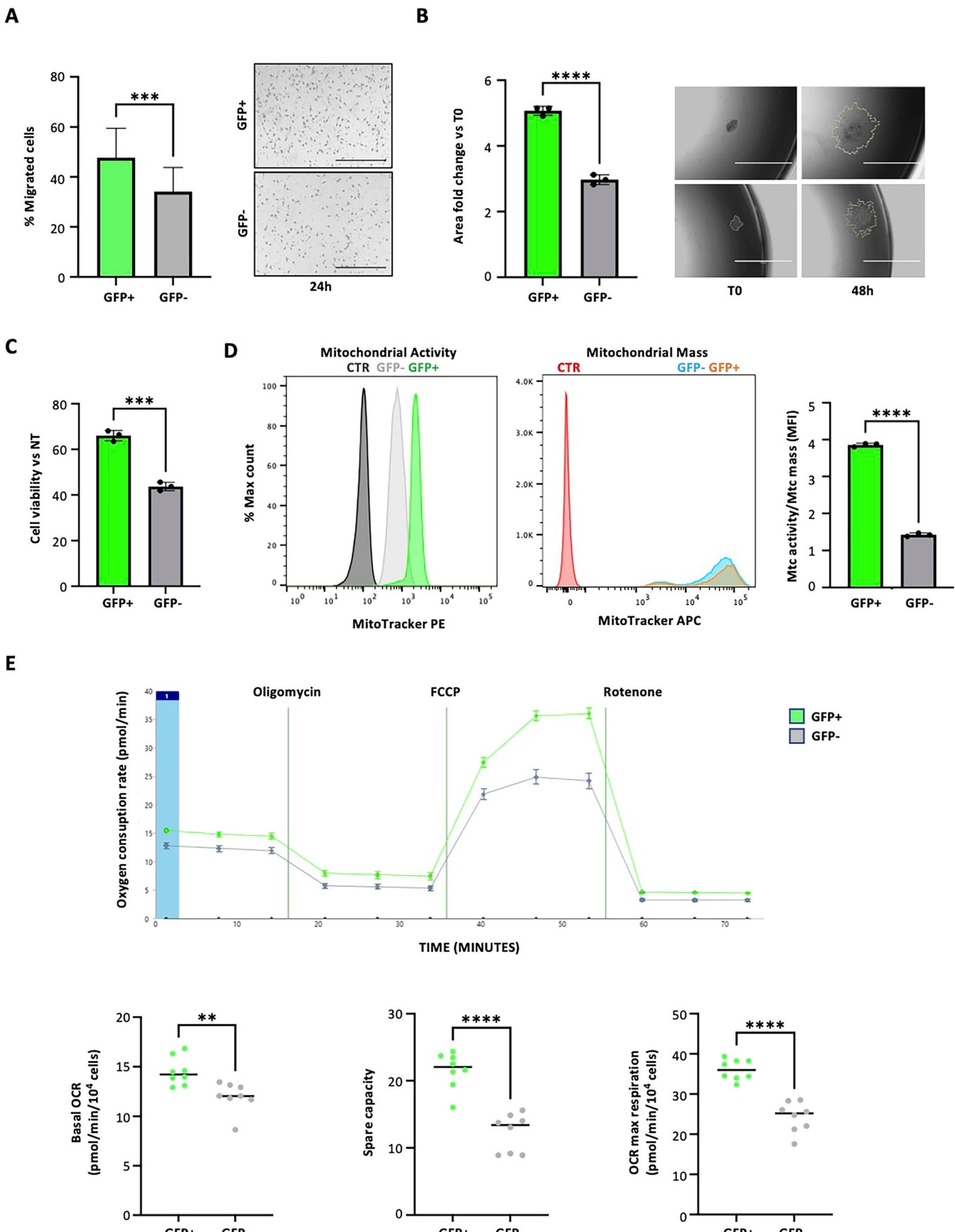

**Figure 4. In vitro functional characterization of MM13 label-retaining cells.**

(A) GFP+ and GFP− MM13 cell migration assessed by transwell migration assay at 24 h. Representative EVOS microscopy images show migrated cells on the transwell outer surface. Data are mean + −SD ($n = 6$ biological replicates). *P* values are based on unpaired Student's *t* test (***$P = 0.000540$). Scale bar $= 400 \, \mu m$. (B) Spheroid collagen invasion area in GFP+ and GFP− MM13 cells measured as fold change vs T0 (imaged daily by EVOS microscopy for 48 h). Data are shown as the mean ± SD of three different spheroids per group ($n = 3$ biological replicates). Student *t* test (****$P = 5.65e\text{-}5$). Representative images are shown. Scale bar $= 1000 \, \mu m$. (C) Treatment of MM13 GFP+ and GFP− cells with 10 nM trametinib for 72 h in vitro. Cell viability assessed by CyQuant and normalized to DMSO control. Mean ± SD ($n = 3$ technical replicates). *P* values are based on unpaired Student's *t* test (***$P = 0.000203$). (D) GFP+ and GFP- MM13 cell mitochondrial activity assessed by flow cytometry analysis of MitoTracker Orange (25 nM) stained cells normalized against mitochondrial mass measured in MitoTracker Deep Red stained cells. Data are shown as ratio of mitochondrial activity/mitochondrial mass mean fluorescence intensity ± SD ($n = 3$ technical replicates). Student *t* test (****$P = 3.19e\text{-}07$). (E) Basal oxygen consumption rate (OCR), OCR spare capacity and OCR max respiration of GFP+ and GFP− MM13 cells assessed by Seahorse XF Cell Mito Stress. Data are mean ± SD ($n = 8$ technical replicates). Student *t* test (**$P = 0.003607$; ****$P = 1.05e\text{-}05$; ****$P = 3.54e\text{-}06$). Source data are available online for this figure.

trametinib, a MEK inhibitor used in the clinic to treat melanoma (Fig. 4C; Appendix Fig. S5C). Furthermore, comparison of the mitochondrial respiratory activities of freshly isolated MM13 and MM27 GFP+ and GFP− cells revealed increased mitochondrial respiration in GFP+ cells, with no change in mitochondrial mass (Fig. 4D; Appendix Fig. S5D). This finding was supported by significantly higher basal and maximal oxygen consumption rates (spare capacity), as well as ATP production measured by the oxygen consumption rate (OCR) in GFP+ cells (Fig. 4E; Appendix Fig. S5E).

Thus, GFP+ cells exhibit enhanced migration, invasion, and drug resistance, suggesting a greater propensity for dissemination and recurrence. Consistently, they displayed increased oxidative phosphorylation and mitochondrial activity, which may allow them to better adapt to microenvironmental changes.

Activation of hypoxia responses is one of the distinguishing features of the transcriptional reprogramming of quiescent cells (Fig. 3B). Tumor tissues experience fluctuating oxygen levels, that arises from dysfunctional tumor vasculature and uneven blood supply, leading to sporadic re-oxygenation periods, and which is crucial in modulating tumor plasticity (Muz et al, 2015; Qiu et al, 2017; O'Connell et al, 2013). In melanoma, low oxygen (LO) conditions drive the ability of melanoma cells to switch between different states, to develop therapy resistance and to exhibit increased expression of markers associated with cancer stemness, particularly those related to slow-cycling properties (Roesch et al, 2010; Fattore et al, 2020; Muz et al, 2015; O'Connell et al, 2013; Cheli et al, 2012). We investigated how hypoxia specifically affects the quiescent melanoma population. Notably, low oxygen (3% $O_2$) increased migration (Fig. 5A; Appendix Fig. S6A), invasiveness (Fig. 5B; Appendix Fig. S6B) and drug resistance (Fig. 5C; Appendix Fig. S6C) of FACS-sorted MM13 and MM27 GFP+ cells, as compared to normal oxygen conditions, while it exerted no or marginal effects on GFP− cells.

We then investigated the effects of different oxygen concentrations on migration and invasion of GFP+ vs GFP− cells, using an in vitro assay that allows variations of oxygen concentration from 0 to 20% in culture (Fig. 5D), mimicking the oxygen fluctuations of the tumor tissue (i.e., from 0 to 9%) (Bertout et al, 2008). MM13 and MM27 GFP+ and GFP− cells were analyzed by time-lapse confocal microscopy for their ability to migrate in 2D (Fig. 5E; Appendix Fig. S6D) and invade collagen matrix in 3D (Fig. 5F; Appendix Fig. S6E) under controlled oxygen conditions. Results demonstrated different propensities of cells to move along oxygen gradients toward hypoxic conditions, with GFP+ populations showing improved migratory and invasive properties in hypoxic conditions in the first 8–16 h of the

experiments (Fig. 5E,F; Appendix Fig. S6D,E). Analyses of cell speed i~n the invasion assay showed an increased average speed of GFP+ cells up to 2–3 h in low oxygen conditions (Fig. 5G; Appendix Fig. S6F). Our data, gathered in a microenvironment that closely mimics the patient's tumor niche, underscores the high capacity of the quiescent population to adapt to low oxygen conditions and acquire invasive properties.

Next, we assessed the tumorigenic properties of purified GFP+ and GFP− cells from MM13 and MM27 PDXs. Cells were injected intradermally into NSG mice to monitor tumor development and metastatic potential upon resection. Both GFP+ and GFP- cells formed primary tumors (Fig. 6A; Appendix Fig. S7A), recapitulating the heterogeneity observed in the original tumors. Notably, tumors derived from both GFP+ and GFP− cells were composed of equal proportions of KI67+ and KI67− cells (Fig. 6B). However, for both PDX models, tumors derived from GFP+ cells exhibited slower growth kinetics and significantly longer latencies, as measured by either time to palpability (Fig. 6C; Appendix Fig. S7B) or time to tumor resection (the latter performed when tumors reached 300 mm³ in volume) (Fig. 6D; Appendix Fig. S7C). Notably, despite only showing a trend towards increased total metastatic burden in various organs (Fig. 6E; Appendix Fig. S7D; Appendix Table S1), GFP+ cells exhibited a significantly shorter latency to lymph node invasion (Fig. 6F; Appendix Fig. S7E). This was accompanied by enlarged lymph nodes (Fig. 6G; Appendix Fig. S7F) and larger metastases within organs, as evidenced by the increased number of human CD298+ cells quantified upon sacrifice (day 100 post injection) (Fig. 6H; Appendix Fig. S7G). In conclusion, tumors derived from GFP+ cells displayed decreased growth rates and increased metastatic potential.

Taken together, our study demonstrates that GFP+ cells isolated from primary human melanomas exhibit enhanced migratory, invasive, and drug-resistant capabilities in vitro, as well as increased metastatic potential in vivo. Considering the higher prevalence of quiescent cells within the GFP+ population, our findings suggest a strong association between quiescence and a heightened capacity for dissemination and recurrence in melanoma. Consistently, our quiescence gene signature proved to be a valuable prognostic biomarker in melanoma patients undergoing targeted or immune therapies.

## Disseminated quiescent cells exhibit reinforced quiescence and transcriptional reprogramming

We next investigated the fate of GFP+ cells generated in the primary tumors (see Fig. 2) during the metastatic process.

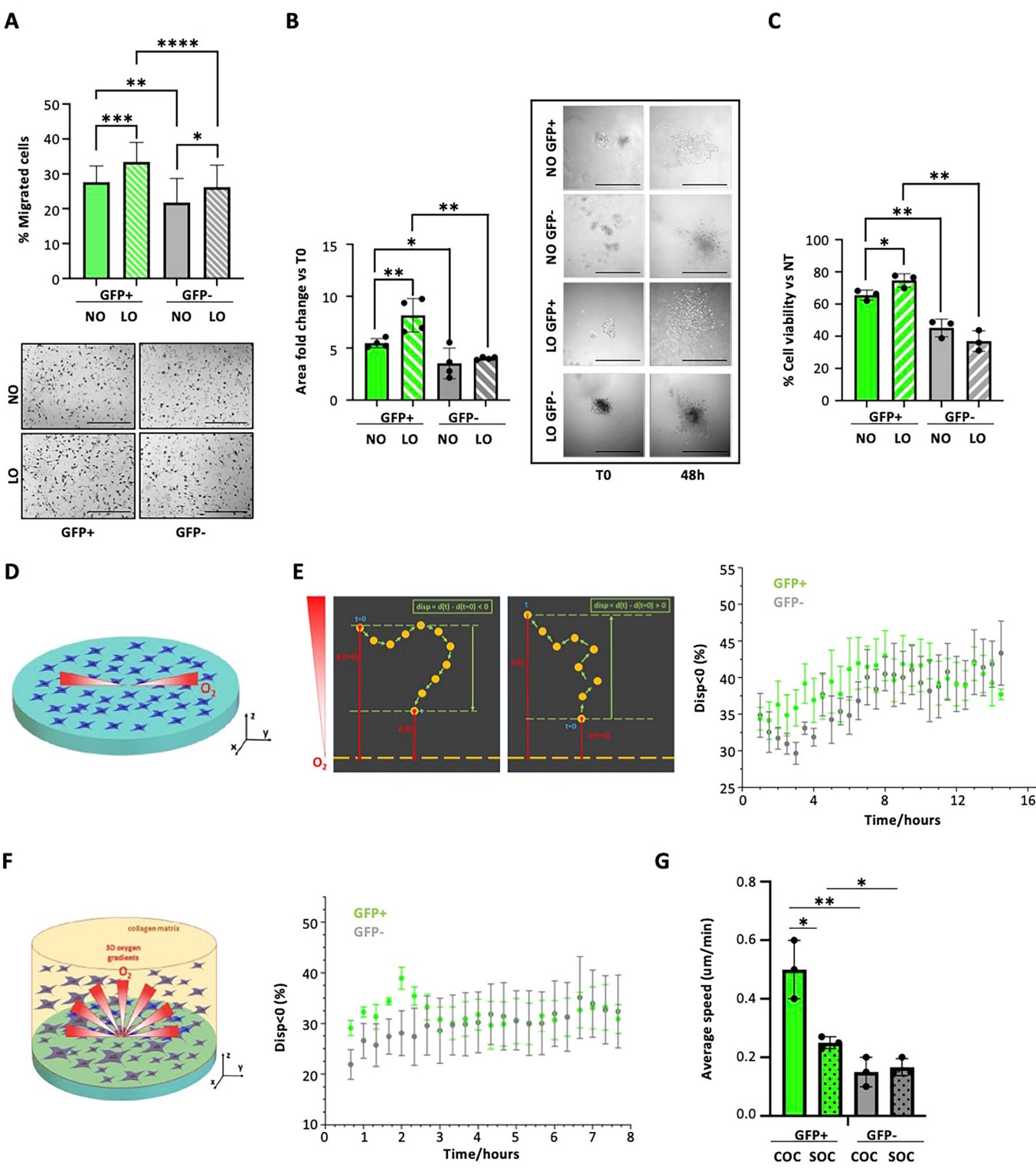

Consistent with previous reports (Perego et al, 2018; Puig et al, 2018; Kusienicka et al, 2023), we identified a small population of GFP+ cells also within lymph node metastases, at frequencies (~2%) significantly higher then in primary tumors (~0.5%) (Fig. 7A). Notably, when isolated from the metastatic sites, GFP+ cells had undergone a longer chasing period (6–7 weeks), which in the primary tumor led to GFP+ complete disappearance (Appendix Fig. S1G), suggesting that the metastatic niche (lymph nodes) supports the quiescence state and promotes long-term survival of disseminated GFP+ cells. Consistently, we observed enrichment of QQ cells in lymph node metastatic samples compared to matched primary tumors (Fig. 7B).

**Figure 5. Impact of the hypoxic tumor microenvironment on MM13 label-retaining cells.**

(A) GFP+ and GFP− cell migration assessed by transwell migration assay after 24 h in normal (NO: 20% O$_2$) and low oxygen (LO: 3% O$_2$) conditions. Representative EVOS microscopy images show migrated cells on the transwell outer surface. Data are mean + -SD ($n = 22$ biological replicates). $P$ values are based on unpaired Student's $t$ test (*$P = 0.0260$; **$P = 0.0021$;***$P = 0.0004$; ****$P = 9.68e\text{-}5$). Scale bar = 400 μm. (B) Spheroid collagen invasion in GFP+ and GFP− cells in normal (20%) and low (3%) oxygen conditions measured as area of invasion fold change vs T0, imaged daily by EVOS microscopy for 48 h. Data shown are the mean ± SD of spheroids ($n = 4$ technical replicates) per group. Student $t$ test (*$P = 0.0444$; **$P = 0.00186$; 0.0022). Representative images are shown. Scale bar = 1000 μm. (C) In vitro 10 nM trametinib treatment of GFP+ and GFP- MM13 cells in normal (20%) and low (3%) oxygen conditions for 72 h. Cell viability was assessed by CyQuant and normalized to DMSO control. Mean ± SD ($n = 3$ biological replicates). $P$ values are based on unpaired Student's $t$ test (*$P = 0.0390$; **$P = 0.0010$; 0.0051). (D) Schematic representation of the experiment conducted under spatially controlled oxygen conditions. (E) Definition of displacement, *disp*, as function of distance of cell position at time $t$, $d(t)$, from zero oxygen level and % of disp <0 (% of cells migrating toward hypoxia) in GFP+ (green) and GFP- (gray) MM13 assessed during time-lapse of migration assay in chemical-controlled culture system. Data are mean ± SD ($n = 30$ technical replicates). Significance was assessed by one-way ANOVA ($P = 0.02298$ and confidence level >95%). Number of trajectories analyzed (GFP+ $n = 668$; GFP− $n = 1158$). (F) Schematic representation of 3D invasion assays in controlled oxygen conditions and percentage of *disp* < 0 in GFP+ (green) and GFP− (gray) MM13 assessed during time-lapse of invasion assay. Data are mean ± SD ($n = 22$ technical replicates). Significance assessed by one-way ANOVA ($P = 0.02587$ and confidence level >95%). Number of trajectories analyzed (GFP+ $n = 1601$; GFP− $n = 976$). (G) Average speed of GFP+ and GFP− cells in 3D collagen invasion assay analyzed in controlled (COC) and in standard (SOC) oxygen conditions. Data are mean ± SD ($n = 3$ biological replicates). Student $t$ test was applied to assess the significance (*$P = 0.013199$ and 0.014732; **$P = 0.005608$). Source data are available online for this figure.

To elucidate how the microenvironment affects the transcriptome and supports quiescence, we performed UMAP clustering and analyzed KI67 expression of GFP+ and GFP− cells from primary melanomas and their corresponding metastases (Fig. 7C). Analysis of cell distribution in the UMAP space revealed significant transcriptional differences between GFP+ and GFP− cells isolated from primary or metastatic sites (Fig. 7C, left panel). Furthermore, analysis of KI67 expression showed a significant decrease in the metastatic GFP+ cells, as compared to GFP+ cells of the primary tumors (Fig. 7C, right panel), suggesting that the metastatic environment enforces a more quiescent state on the GFP+ population. Clustering and UMAP analysis of both primary tumors and metastasis QQ and PQ cells revealed a striking difference in cell distribution across clusters between the two samples, with the identification of clusters unique to each population (Fig. 7D for QQ; Appendix Fig. S8A for PQ). Metastatic quiescent cells exhibited marked enrichment in clusters 2, 3, 4, 7, 8 in QQ (Fig. 7E) and 0, 5, 6 in PQ states (Appendix Fig. S8B). Thus, the quiescent cells undergo transcriptional reprogramming upon homing to the metastatic niche, suggesting that metastatic environment shapes the gene expression profile of these quiescent cells.

Trajectory analyses confirmed the different transcriptional profiles of quiescent cells in primary tumors and corresponding metastases (Fig. 7F for QQ; Appendix Fig. S8C for PQ). Indeed, unsupervised trajectory analysis identified 5 different states in both QQ (Fig. 7F,G) and PQ (Appendix Fig. S8C,D) cell populations with primary QQ cells enriched at early stages of the trajectory, while metastatic QQ cells enriched at later points of pseudotime (Fig. 7F,G). Interestingly, the quiescent PQ cells showed a similar trend but with less degree of enrichment between early and later stage in primary and a bigger degree of enrichment between early and later stage in metastasis (Appendix Fig. S8D). These data confirm that melanoma quiescent cells undergo transcriptional reprogramming during melanoma progression.

To understand the unique transcriptional program of the QQ melanoma cells, we compared their gene expression profiles to those of all other cell populations within primary tumors or metastases. Our analysis revealed distinct sets of genes enriched in QQ cells, 321 markers specifically upregulated in primary tumors and 752 in metastases (Dataset EV2). By focusing on the genes expressed in both groups, we identified a core set of 189 genes that form a unique "melanoma quiescence signature" (Fig. 7H, left

panel). Using GSEA analysis, we compared markers specifically associated with metastatic and primary QQ cells to the rest of the population. This analysis showed distinct biological programs associated with quiescence in different locations. In primary tumors, quiescence was linked to pathways involved in cellular iron regulation, stress response, and apoptosis. Conversely, quiescence in metastases was associated with pathways involved in epithelial to mesenchymal transition, regulation of angiogenesis, Wnt pathway and response to growth factors (Fig. 7H, right panel). Both QQ primary and metastatic cells rely on the upregulation of pathways involved in cellular response to oxidative stress, regulation of apoptotic process and mitochondrial membrane permeability (Fig. 7H, right panel). This signature highlights a potential link between the quiescent state and specific cellular programs, as several top-ranked markers (PRRX1, LUM and SPARC) significantly overlapped with a previously described signature for a "mesenchymal-like" state of melanoma metastatic initiating cells (Karras et al, 2022) (Appendix Fig. S9A). Interestingly, the genes associated with the PRRX1 regulatory network, were significantly upregulated in metastatic QQ cells (Appendix Fig. S9A), suggesting that the metastatic environment enforces the mesenchymal phenotype (potentially through PRRX1) and disrupt cell- extracellular matrix interactions (potentially through LUM and SPARC) as hallmarks of the quiescent state.

These findings demonstrate that the metastatic microenvironment reprograms the gene expression profiles of disseminated melanoma cells and promotes their dormancy. This reprogramming might equip these cells with novel functionalities, such as the mesenchymal phenotype, potentially contributing to their long-term survival.

## GPNMB is a specific marker of the quiescent state

The data described above strongly suggest that the pro-metastatic phenotype of GFP+ cells is linked to the prevalence of quiescent cells within the GFP+ population, as defined by the negativity of KI67 expression. Unfortunately, however, KI67 is a nuclear antigen and cannot be used to purify viable quiescent cells to directly test their biological properties. To circumvent this limitation, we analyzed the transcriptional characteristics of KI67+ versus KI67- cells, searching for surrogate surface markers that could be used for their purification.

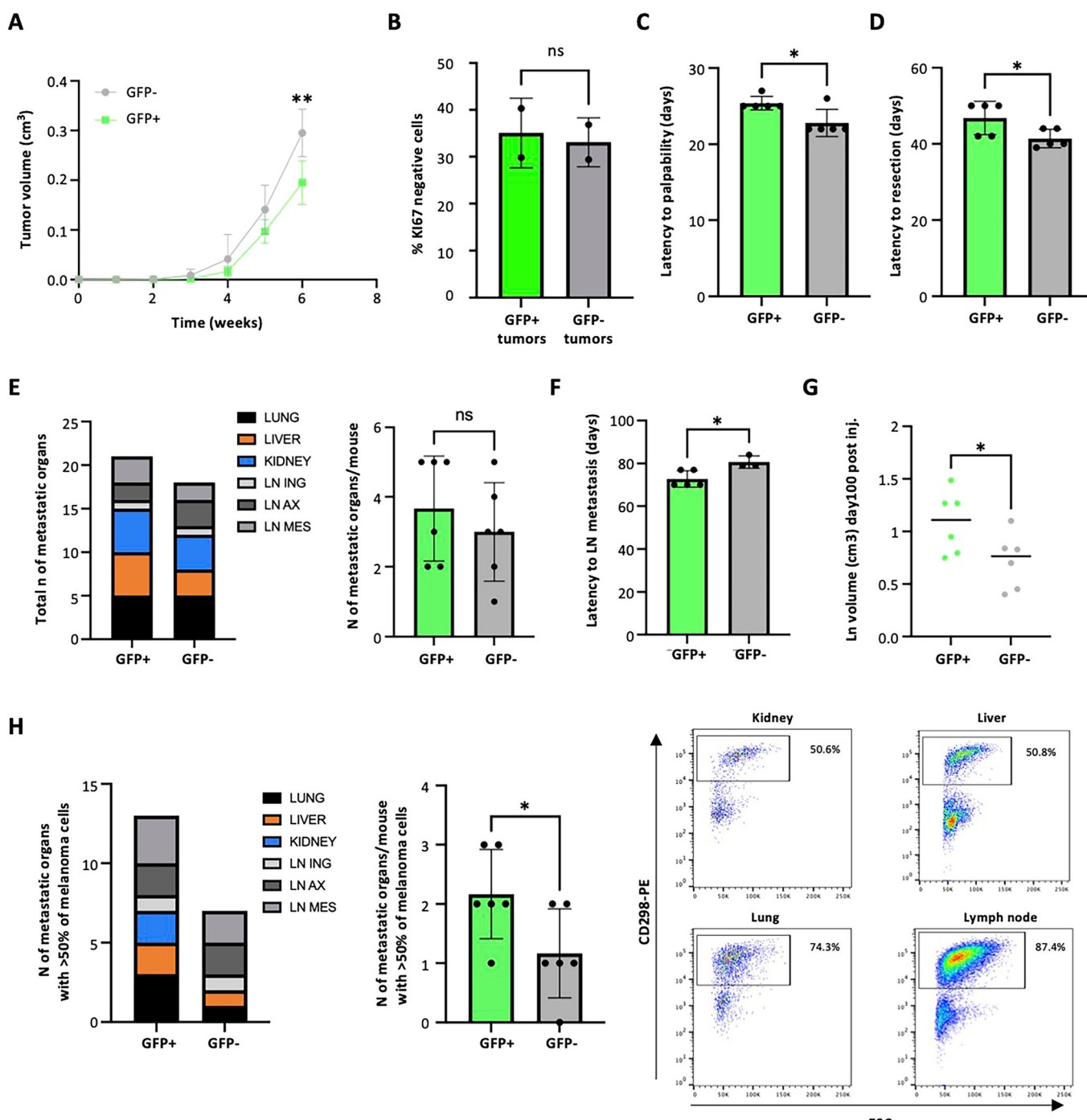

We sorted MM13 cells based on KI67 protein expression (KI67+ and KI67− cells), performed bulk RNA-seq analysis and identified 424 upregulated and 1079 downregulated genes in KI67- compared to KI67+ cells ($|\log2FC| > 1$, $P$ value < 0.01) (Appendix Fig. S9B). Comparing the expression of genes upregulated in KI67− cells with marker genes of the QQ cell state (Dataset EV1: Appendix Fig. S9B), we found 77 upregulated genes (18%) in common (Appendix Fig. S9C), that further confirm the specificity of the newly identified melanoma quiescent cells signature. We then focused on the surface glycoprotein-NMB (GPNMB) as the

first ranked gene in the list and the first actionable target within the quiescence signature (Fig. 8A). GPNMB was highly expressed in QQ and PQ cells compared to other cell states (Figs. 8B and 3A). Further supporting this link, UMAP clustering based on GPNMB expression (75th percentile) effectively separated quiescent cells from all other cell types (Fig. 8C). Moreover, GPNMB expression exhibited a significant inverse correlation with KI67 (Fig. 8D), with higher levels observed in the quiescent cell states (QQ and PQ cells) compared to their proliferating counterparts (PP and PQ cells) (Fig. 8E). Consistently, GPNMB+ cells were predominantly KI67−

◄ **Figure 6. In vivo assessment of tumorigenic and invasive properties of GFP+ and GFP- MM13 populations in immunocompromised mice.**

(A) Tumor growth at 6 weeks. Data are mean ± SD ($n = 5$ biological replicates). Student t test was used (**$P = 0.0089$). (B) Cells derived from GFP+ and GFP− MM13 tumors stained with KI67 and analyzed by flow cytometry. Data are presented as mean ± SD ($n = 3$ biological replicates). Student t test was used (ns = 0.7901, not significant). (C) Latency to tumor palpability. Data are presented as mean ± SD ($n = 5$ biological replicates). Student t test was used (*$P = 0.0197$). (D) Latency to tumor resection (volume ~0.3 cm³). Data are presented as mean ± SD ($n = 5$ biological replicates). Student t test was used (*$P = 0.0422$). (E) Total number of organs with detectable metastases (lung, liver, kidney, inguinal lymph node LN ING, axillary lymph node LN AX, mesenteric lymph node LN MES) and average number of organs with metastases per mouse. Data are presented as mean ± SD ($n = 6$ biological replicates). Student t test was used (ns = 0.4475, not significant). (F) Latency to lymph node (LN) metastasis (days). Data are presented as mean ± SD ($n = 5$ biological replicates). Student t test was used (*$P = 0.0229$). (G) LN volume at day 100 after injection. Data are presented as mean ± SD ($n = 6$ biological replicates). Student t test was used (*$P = 0.0475$). (H) Total number of organs with at least 50% of melanoma cells and average number of organs with at least 50% of melanoma cells per mouse. A representative flow cytometry analysis is reported to show the assessment of metastatic cells within each organ based on the expression of human CD298. Data are mean ± SD ($n = 6$ biological replicates). Student t test was used (*$P = 0.0442$). Source data are available online for this figure.

(81.5%), as determined by immunofluorescence analyses of MM13 primary tumors (Fig. 8F; Appendix Fig. S9D). To provide important information regarding the spatial context of our quiescent cells and their relationship to hypoxic regions within the tumor microenvironment, which may contribute to their functional differences, we spatially analyzed the distribution of GPNMB-positive quiescent cells (QQ and PQ) within hypoxic areas of primary MM13 tumors, as defined by pathological examination. Indeed, GPNMB-positive cells preferentially localize in the peri-hypoxic areas with a 2.3-fold increase (14.8%) compared to the normoxic areas (6.5%) (Appendix Fig. S9E). Intriguingly, GPNMB expression was not only associated with the quiescent state but also with the metastatic phenotype. Indeed, the percentage of GPNMB + /KI67− cells rises to 91.3% in the lymph node metastases (Fig. 8F; Appendix Fig. S9D). Moreover, GPNMB was upregulated in metastatic QQ cells, as compared to their primary counterparts (Fig. 8G), and, among the "growth inhibitory genes" of the melanoma-metastasis gene MODULE_488, it emerged as the most significantly upregulated gene specifically in metastatic quiescent cells (Appendix Fig. S9F).

To validate these findings across a larger cohort of patients, we analyzed data from The Cancer Genome Atlas (TCGA). Consistent with our observations, GPNMB expression was significantly higher in the KI67- quiescent melanoma cells isolated from metastatic tumors compared to quiescent cells from primary tumors (Fig. 8H). Together, these findings suggest that GPNMB expression marks the quiescent state of melanoma cells and is also associated with their metastatic potential, highlighting GPNMB as a marker to isolate quiescent cells in melanomas, and potential target of the metastatic process in melanoma.

## GPNMB-expressing melanoma cells exhibit a pro-metastatic differentiated phenotype

To investigate functional properties of GPNMB-expressing cells, MM13 GPNMB+ cells were FACS-sorted using a specific antibody against the extracellular domain of GPNMB and analyzed for their invasive and drug resistance properties. GPNMB+ cells showed increased invasiveness (Fig. 8I) and resistance to trametinib treatment (Fig. 8J). Consistently, GSEA analysis performed on scRNA-seq of GFP + GPNMB+ vs GFP-GPNMB− populations revealed enrichment of pro-invasive and pro-tumorigenic pathways, such as extracellular matrix remodeling, microenvironment acidification and response to oxygen (Appendix Fig. S9G). Gene expression analysis showed that GPNMB+ cells significantly upregulated genes involved in invasion and/or tissue remodeling, including EGFR, a direct interactor of GPNMB in breast cancer cells (Lin et al, 2016), its downstream targets (STAT3, MMP 2-3-9) and ITGB4, TGFβ, VEGF and VCAM (Appendix Fig. S9H). Moreover, GPNMB+ cells showed a differentiated phenotype, as indicated by the upregulation of markers of melanocytic lineage, such as tyrosinase (TYR), MITF and ABCB5, and the down-regulation of AXL gene expression (Appendix Fig. S9I).

To test their tumorigenic and metastatic potential, we injected intradermally 10,000 GPNMB+ or GPNMB− cells derived from MM13 (Fig. 8K–M) and MM27 (Appendix Fig. S9J–L) PDX models. Compared to GPNMB-, GPNMB+ cells exhibited significantly longer latency of tumor formation, measured by either time to palpability (Fig. 8K, Appendix Fig. S9J) or time to tumor resection (volume of ~0.3 cm³) (Fig. 8L; Appendix Fig. S9K) but earlier latency of lymph node invasion (Fig. 8M; Appendix Fig. S9L and Appendix Table S2), consistent with their less proliferative and more invasive phenotype.

To target GPNMB+ cells we employed glembatumumab vedotin (GV)(Rose et al, 2017), an antibody–drug conjugate (ADC) that binds the extracellular domain of GPNMB (Rose et al, 2017). As expected, in vitro treatment with GV resulted in massive cell death of GPNMB+ cells, with minimal effect on GPNMB- cells (Fig. 8N). In vivo, GV treatment of mice bearing MM13 tumors did not significantly alter tumor latency (Fig. 8O) while dramatically reducing the metastatic potential, as evidenced by a significant decrease in the number of organs bearing metastases and the absence of detectable metastatic lymph nodes in treated mice (Fig. 8P,Q; Appendix Table S3).

The effective block of the metastatic spread provides substantial evidence that cells expressing GPNMB represent the metastatic population. This is also a clear and compelling demonstration that quiescent cells exhibit a higher metastatic potential compared to their active proliferating counterparts.

Collectively, these data demonstrate that while GPNMB+ cells promote invasiveness and drug-resistance, they contribute to a less proliferative state, leading to a paradoxical phenotype with delayed tumor formation. Nevertheless, GPNMB+ cells appear to be crucial for melanoma dissemination. Thus, GPNMB+ cells recapitulate the biological properties of GFP+ cells, providing a direct evidence for the role of quiescence in driving the pro-metastatic phenotype. Our findings also demonstrate the potential of targeting GPNMB with GV as a therapeutic strategy to prevent metastatic spread in melanoma patients.

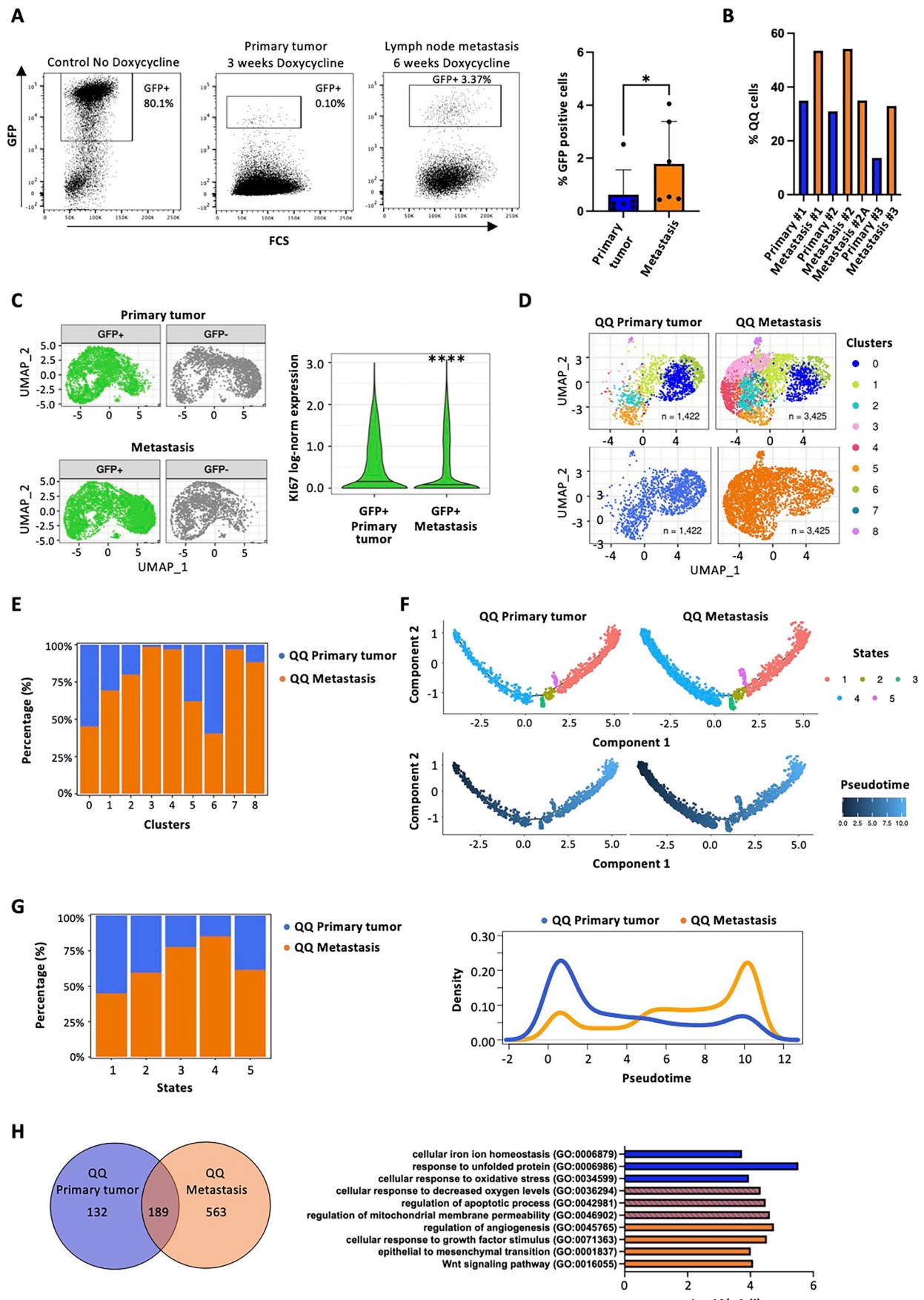

**Figure 7.  Analysis of transcriptional heterogeneity of MM13 melanoma quiescent QQ cells.**

(A) FACS analysis (representative dot plot and bar plot) showing the percentage of GFP+ cells in primary tumors and matched metastatic lymph nodes. Mice ($n = 6$ biological replicates) are treated with doxycycline for 3 weeks while primary tumors are growing, followed by tumor resection. Doxycycline treatment was continued for an additional 3 weeks until lymph node metastasis formation. Student $t$ test was applied to assess the significance (*$P = 0.0312$). (B) Bar plot showing percentage of QQ cells in each primary tumor and matched metastatic lymph-nodes (Prim $n = 3$ and Met $n = 4$ biological replicates: Prim/Met#1: 34.9%/53.5%; Prim#2/Met#2,2A: 30.9%/ 54.2%, 35%; Prim/Met#3: 13.7%/32.9%). (C) UMAP projection of GFP+ and GFP− compartments of MM13 melanoma primary and metastatic populations (left); violin plot of KI67 expression in GFP+ compartment of MM13 melanoma primary and metastatic populations (right). Mann–Whitney test was applied to assess the significance ($P**** = 1.29e-29$). (D) UMAP projection of the analysis of re-clustered QQ cells derived from primary tumor and metastatic samples. (E) Stack plot showing relative proportions of primary and metastatic QQ cells across Seurat clusters. Percentages are calculated as the proportion of QQ primary or QQ metastatic cells within each cluster, relative to the total number of cells in that specific cluster. (F) Trajectory analysis of primary and metastatic QQ cell populations based on the expression of the top 20 variable genes. The trajectories are colored by state (upper panel) and pseudotime (lower panel). State 1 was selected as root state. (G) Stack plots (left) showing relative proportions of primary and metastatic QQ cells across pseudotime states; density plots (right) showing distribution of primary and metastatic QQ cells across along pseudotime. (H) Venn diagram (left) of marker genes of QQ populations in primary tumors and metastases (Dataset EV2); GSEA functional annotation (right) of 189 common genes. Test with FDR correction was applied. Source data are available online for this figure.

# Discussion

Cellular quiescence is a state of reversible growth arrest in which cells have exited the cell cycle, yet they can re-enter it upon appropriate stimulation (Sistigu et al, 2020). Quiescent cells play a key role in the physiology of tissue homeostasis and represent a major hurdle in cancer treatment. Indeed, quiescent cells are resistant to anti-cancer therapies and they can remain in a dormant state for long periods of time, reawakening to give rise to cancer recurrence and metastasis (Lindell et al, 2023). Thus, they represent a critical target population to prevent relapse in different tumors, including melanoma (Janowska et al, 2022). Elucidating molecular mechanisms of quiescent melanoma cells, in both primary melanomas and as they spread, is crucial to unlocking the complex mechanisms underlying melanoma progression. Here we combined the H2B-GFP label-retention system and a surface marker approach to isolate and extensively characterize in vitro and in vivo quiescent cells in our NRAS or BRAF mutated human melanoma PDXs. These two PDXs faithfully recapitulate the patient genetic and phenotypic landscape (Bossi et al, 2016) and represent excellent models of spontaneous metastasis when injected intradermally.

To investigate the dynamics of cell proliferation during the H2B-GFP label-retention assay, we analysed GFP+ cells using ProCell, a novel computational framework which takes into account the inherent stochasticity of cell cycle entry (Nobile et al, 2019; Nobile et al, 2019). By analyzing experimental data from our doxycycline-induced PDX primary tumors, ProCell accurately modeled the diverse proliferation rates of GFP-labelled melanoma cells, predicting the existence, in the whole tumor population at steady state, of ~30% quiescent and a 70% proliferating cells. Previous studies have interpreted the results of label-retaining experiments as indicative of the existence in melanomas of a small-sized subpopulation of slow-cycling cells (Roesch et al, 2010; Roesch et al, 2013; Perego et al, 2018; Puig et al, 2018).

We have shown that both the label-retaining GFP+ and the label-negative GFP-cells are composed of co-existing KI67-/BrdU− and KI67 + /BrdU+ cells, though at different relative proportions (70–30% and 30–70%, respectively). This data suggest that GFP+ quiescent cells can re-enter the cell-cycle, while GFP- proliferating cells can become quiescent, e.g. that quiescence is a functional state that mark all melanoma cells.

This hypothesis was supported by gene-expression analyses of GFP+ and GFP- cells. In order to identify the population of melanoma quiescent cells within the GFP label-retaining population, we performed single-cell RNA-sequencing (scRNA-seq) analysis and stratified cells according to the median of KI67 mRNA expression. We identified for the first time four transcriptionally distinct cell states, the GFP label-retaining quiescent (QQ), the GFP label-retaining proliferating (QP), the GFP label-negative quiescent (PQ), and the GFP label-negative proliferating (PP) cells. Importantly, transcriptional profiles of the quiescent QQ and PQ cells were largely similar and distinguished from those of the cycling QP and PP cells, and occupied peculiar functional states in the pseudotime analysis.

GSEA showed that QQ and PQ cells activate pathways involved in the regulation of energy-depletion, stress responses and cell cycle inhibition, while the proliferating QP and PP states showed enrichment of genes associated with cell cycle progression. Of note, our single cell transcriptomic data suggest that quiescent QQ cells display specific modulation of iron ion homeostasis and oxygen metabolism, which promote invasive behavior. While hypoxia conditions are known to play a role in pushing cells towards quiescence (Roesch et al, 2010; Fattore et al, 2020; Muz et al, 2015; O'Connell et al, 2013; Emami Nejad et al, 2021), the iron ion homeostasis has been linked to cancer stemness maintenance (Katsura et al, 2019; Wang et al, 2022; Guo et al, 2021). Furthermore, both oxidative stress and iron metabolism are linked to ferroptotic process, suggesting that quiescent melanoma cells could be sensitive to ferroptosis inducing agents (Perego et al, 2018). Iron overload, antioxidant inhibitors or combination of ferroptosis inducers with traditional therapies can disrupt the cellular antioxidant defense system, providing a new therapeutic approach for targeting metastatic quiescent cells to prevent relapse, especially in the still under explored adjuvant setting (Ta et al, 2023; Zhou et al, 2024).

Our data also allowed, for the first time, to characterize quiescent cells from primary tumors and metastasis. Re-clustering of quiescent cells based on their origin (primary tumor vs. metastasis), showed striking diversity within the QQ cells. Primary-tumor QQ cells displayed enriched pathways related to iron ion homeostasis and stress response, while metastatic QQ cells showed enrichment in pathways associated with epithelial-to-mesenchymal transition, Wnt signaling, and angiogenesis. This

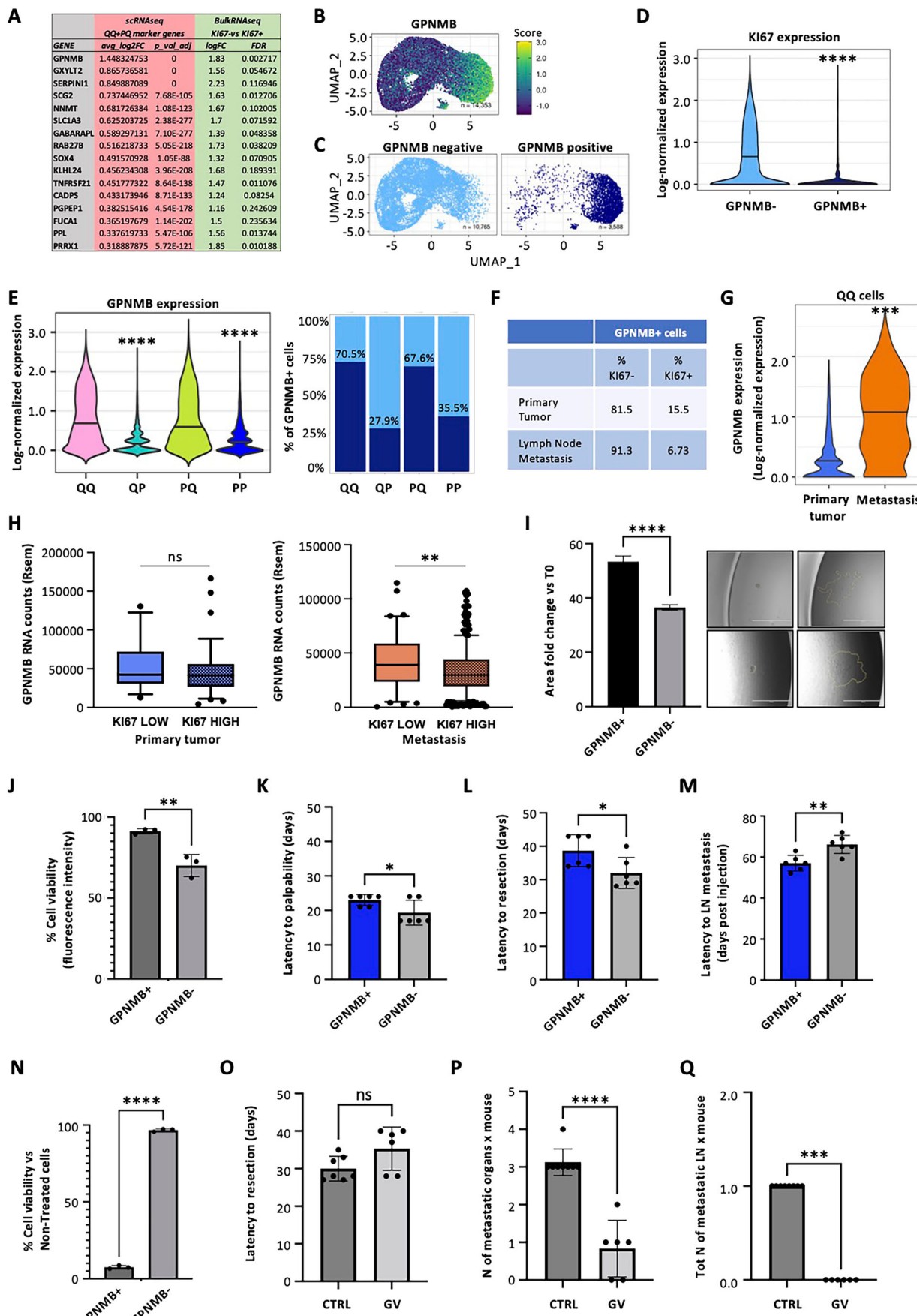

◀

**Figure 8. GPNMB is a master regulator of the quiescent state in primary and metastatic melanoma.**

(A) List of the top 16 genes upregulated in the QQ + PQ signature derived from scRNA seq of primary and metastatic MM13 samples and bulk RNA-seq of KI67- MM13 cells. (B) UMAP projections of MM13 melanoma dataset (Fig. 5A) colored by GPNMB expression. (C) UMAP projection of MM13 melanoma dataset grouped according to 75th percentile of GPNMB expression. (D) Violin plot of KI67 expression in MM13 GPNMB− and GPNMB+ populations (Mann–Whitney test, ****$P = 2.2e$-16). (E) Violin plot (left panel) and stack plot (right panel) of GPNMB expression in MM13 PP-PQ-QP-QQ populations. (Mann–Whitney test, ****$P = 2.2e$-16). (F) Table showing the percentage of KI67- and KI67+ cells within the MM13 GPNMB+ population of primary tumor and lymph node metastasis ($n = 1$). (G) Violin plot of GPNMB expression in primary and metastatic QQ cell populations. (Mann–Whitney test, ***$P = 0.000145$). (H) GPNMB expression in the TCGA melanoma datasets of primary (blue) and metastatic (brown) patients stratified on the 25th percentile KI67 expression. (Mann–Whitney test, $P$ ns = 0.246 not significant; **$P = 0.025$). Primary patients KI67 LOW ± SD28036: min 12496, max 130359, median 42467, 25%: 30745, 75%: 71836; Primary patients KI67 HIGH ± SD 28609: min 4200, max 166506, median 41492, 25%: 26742, 75%: 56080; Metastatic patients KI67 LOW ± SD 23830: min 449, max 114688, median 39158, 25%: 23436, 75%: 58891; Metastatic patients KI67 HIGH ± SD22920: min 369, max 107391, median 29674, 25%: 19207, 75%: 44352. (I) Spheroid collagen invasion assay in GPNMB+ and GPNMB− MM13 cells. The area of invasion was monitored and imaged daily by EVOS microscopy for 48 h. Data are shown as the mean ± SD of four different spheroids per group (technical replicates) and Student $t$ test applied (****$P = 6.39e$-6). Representative images are shown. Scale bar = 1000 μm. (J) In vitro 10 nM trametinib treatment of GPNMB+ and GPNMB- MM13 cells for 72 h. Cell viability assessed by CyQuant and indicated over control ($n = 3$ biological replicates). Mean ± SD. $P$ values are based on unpaired Student's $t$ test (**$P = 0.0065$). (K–M) In vivo assessment of tumorigenic and invasive properties of $10^4$ GPNMB+ and GPNMB- MM13 populations injected intradermically in NSG mice ($n = 6$) (Appendix Table S2): (K) latency to tumor palpability ($n = 6$ biological replicates). Student $t$ test was applied to assess the significance (*$P = 0.0455$); (L) latency to tumor resection (volume ~0.3–0.4 cm³) ($n = 6$ biological replicates). Student $t$ test was applied to assess the significance (*$P = 0.0340$); (M) latency to lymph node (LN) metastasis; Data are mean ± SD ($n = 6$ biological replicates). Student $t$ test was applied to assess the significance (**$P = 0.0065$). (N) In vitro 1 μM Glembatumumab vedotin (GV) treatment of GPNMB+ and GPNMB- MM13 cells for 72 h. Cell viability assessed by CyQuant and indicated over control ($n = 3$ technical replicates). Mean ± SD. $P$ values are based on unpaired Student's $t$ test (****$P = 4.95e$-8). (O–Q) In vivo assessment of GV efficacy on MM13 melanoma cells in NSG mice ($2 \times 10^5$ cells injected intradermally) (Appendix Table S3): (O) latency to tumor resection (volume ~0.3–0.4 cm³) (Control $n = 7$ biological replicates; GV $n = 6$ biological replicates). Student $t$ test was applied to assess the significance (ns $P = 0.06$ not significant); (P) total number of organs with detectable metastases per mouse. Data are mean ± SD (control $n = 8$; GV $n = 6$ biological replicates). Student $t$ test was applied to assess the significance (****$P = 6.058e$-6); (Q) total number of metastatic lymph-nodes per mouse. Data are mean ± SD (control $n = 8$; GV $n = 6$ biological replicates). Student $t$ test was applied to assess the significance (***$P = 0.0003$). Source data are available online for this figure.

highlights the functional heterogeneity within the quiescent cell population, with primary and metastatic niches potentially shaping their gene expression profiles. Consistent with single cell transcriptomic findings, we found that QQ cells are specifically enriched in metastases compared to matched primary tumors in all experimental animals. These data imply that the environment within metastases acts as a selective force, promoting the accumulation of tumor cells in a quiescent state. The identified signature of quiescent melanoma cells was further validated by its association with key clinical challenges. Tumors expressing this signature were linked to both increased risk of relapse and resistance to specific therapies, namely MAPK inhibitors and immune checkpoint therapies. Our study suggests that proteins encoded by genes uniquely expressed in QQ cells could be promising new targets for melanoma therapy.

We demonstrated that GFP+ cells display an increased capability of migrating, invading matrix in vitro and tissues in vivo, and are refractory to therapy. Puig et al, previously used a similar approach to define a slow cycling, drug-resistant melanoma population characterized by a distinctive gene expression profile, enriched in pathways related to drug detoxification, stemness, hypoxia, and crosstalk with the immune system (Puig et al, 2018). The authors demonstrated the association of JARID1B expression with slow-cycling melanoma cells, as previously reported (Roesch et al, 2010), and identified their ability to transition between states with or without JARID1B expression, suggesting a more heterogeneous nature than previously thought (Puig et al, 2018). Slow-cycling cells have been identified not only in primary melanomas but also in metastases using PKH26, a membrane dye that allows identification of label-retaining cells incapable of dividing (Nabil et al, 2021). However, no specific phenotype or marker has been definitively associated with slow-cycling cells, highlighting the need for a more nuanced

understanding of these cells in primary melanoma and potentially associated metastases (Roesch et al, 2010; Puig et al, 2018).

As part of a comprehensive characterization of GFP+ cells, we explored their metabolic behavior, showing that these cells favor oxidative phosphorylation (OXPHOS) for energy production within the mitochondria.

Previous research has shown that aspects of the TME, such as hypoxia, acidity, nutrient deprivation, can promote tumor cell quiescence, progression, and resistance to therapy (Bailey Kate et al, 2012). Notably, these pathways are distinguished features of the transcriptomic profile of quiescent cells in our scRNAseq analyses. To gain a deeper understanding of how the TME affects different subpopulations of melanoma cells, we developed a new in vitro model that offers several advantages: it is easy to handle, replicates key features of real tumor tissue, and allows us to precisely control oxygen levels. While "organs on a chip" technology using microfluidics has been used to model tumor tissue, it does not fully capture the intricate network of cells and signaling molecules that constitute the tumor microenvironment (Bhatia and Ingber, 2014; Esch et al, 2015). We therefore used a technology that was developed to modulate oxygen levels in normal culture conditions (Becconi M et al) to investigate the specific functional properties of dynamic-quiescent cells. We demonstrated that GFP+ cells have enhanced migratory and invasive properties and increased cell speed in a setting where oxygen levels can be fine-tuned. Indeed, they migrate rapidly to hypoxic areas, in contrast to GFP− cells. Previous research (Widmer et al, 2013) has established the role of low oxygen environments in promoting metastasis by triggering a switch to a more invasive cell state. Our findings provide a new piece to this puzzle, suggesting that hypoxia may not only fuel invasion but also play a critical role in maintaining the population of quiescent melanoma cells. Indeed, the transcriptional profile of QQ cells shows these cells display an enrichment of genes

associated with the cellular response to low oxygen conditions (Fig. 3B).

To directly prove that the pro-metastatic and drug resistance phenotypes of GFP+ cells were indeed due to their predominantly quiescent state, we characterized the transcriptomic profiles of KI67− vs KI67+ cells searching for surface molecules that could be used as actionable markers of the quiescence state. Our finding pinpoints GPNMB, a protein uniquely expressed by quiescent cells (QQ and PQ), as a potentially groundbreaking therapeutic target for melanoma. This discovery adds another layer of complexity to the mechanisms driving metastasis. The identification of GPNMB in both primary and metastatic quiescent cells highlights its potential as a therapeutic target, warranting further investigation and drug development. GPNMB is a type I transmembrane protein abundantly expressed in melanoma compared to normal tissue and other solid tumors such as glioblastoma and breast cancer where it has been reported as a prognostic factor (Kuan et al, 2006; Maric et al, 2013; Huang et al, 2021b; Feng et al, 2020). It is linked to tumor development, cell movement, invasion, and metastasis, and has recently gained attention as a promising target for cancer treatment (Taya and Hammes, 2018; Arosarena et al, 2018; Oyewumi et al, 2016; Fiorentini et al, 2014). In breast cancer, its expression correlates with dormancy and stemness (Chen et al, 2018; Okita et al, 2018). Moreover, increasing evidence suggests that GPNMB plays a role in suppressing the immune system in cancer, thus promoting tumor progression (Lazaratos et al, 2022). Intriguingly, our analysis identified SPARC, a known immunosuppressive glycoprotein, within the QQ cell signature. Notably, SPARC has previously been linked to acquired tumor cell invasion abilities, a process often facilitated by a suppressed immune response. This finding suggests a potential link between quiescent cells, SPARC, and the immunosuppressive microenvironment that can contribute to melanoma progression, similarly to GPNMB (Arosarena et al, 2018; Oyewumi et al, 2016; Fiorentini et al, 2014; Chen et al, 2012; Morrissey et al, 2016).

Our data show that the expression of GPNMB on the cell surface marks the quiescent phenotype and exhibits invasive and therapy-resistant features together with a significant negative correlation with KI67 levels in metastatic melanoma patients. Understanding its role in melanoma may provide insights into the broader mechanisms of quiescence and metastasis. GPNMB can promote tumor progression in different ways, such as enhanced cancer proliferation, survival, invasion, angiogenesis (Rose et al, 2017). Dissecting a possible mechanism of action, GPNMB has been reported to directly interact with EGFR, thus triggering a signaling cascade on its downstream targets, STAT3 and MMP2-3-9, and on VEGF and VCAM (Lin et al, 2016), suggesting that it may help creating a complex relationship between invading tumor and vasculature during tumor invasion. The quiescent phenotype mediated by GPNMB likely involves multiple mechanisms at the molecular level that need to be further addressed. GPNMB, a marker for cancer stem cells in many cancers, presents a paradox in melanoma. Typically associated with an undifferentiated state, GPNMB has also been linked to cells expressing high levels of MITF associated with a more mature, differentiated state (Rose et al, 2017). This suggests that quiescent cells, regardless of their differentiation status as defined by MITF levels, may exhibit a more aggressive phenotype, including increased invasiveness, drug resistance, and tumor initiation potential. This is reminiscent of

the ABCB5 gene (Schatton et al, 2008; Louphrasitthiphol et al, 2019), where its expression seems to mark slow-cycling, MITF-high tumor-initiating cells with a more differentiated phenotype.

Glembatumumab Vedotin (GV), an antibody–drug conjugate (ADC) targeting GPNMB, is being investigated for treating breast (Vahdat et al, 2021) and lung (Khan et al, 2021) cancers, and in advanced melanoma. Studies include use as monotherapy and in combination with immunotherapy (phase II trial for melanoma) (Ott et al, 2019). While previous trials of Glembatumumab Vedotin (GV) did not achieve all primary endpoints, the drug was well-tolerated with minimal side effects. This suggests that GV may still hold promise for melanoma treatment. Our research strengthens this possibility by identifying GPNMB as a specific marker for targeting the pro-metastatic and drug resistance phenotypes of quiescent melanoma cells. This opens the door to a new approach in clinical trials, focused on the prevention of treatment or drug resistance, rather than treatment of the metastatic disease. Thus, analyses of metastasis-free survival or relapse rates may represent better endpoints in the neo-adjuvant or adjuvant settings. Furthermore, the potential synergy of GV with targeted therapies (MEKi or MAPKi) or EGFR inhibitors combined with immunotherapy warrants further investigation. These findings pave the way for a more targeted and potentially more effective approach to treating melanoma progression.

In conclusion, our research sheds new light on the characteristics of quiescent melanoma cells within primary tumors and metastases, offering valuable clues to unravel the biological mechanisms that drive and maintain dormancy in melanoma cells. The most compelling data comes from GPNMB-targeted ADC, which effectively blocks metastases and provide strong evidence that GPNMB-quiescent cells represent the metastatic population. Building on this knowledge, future studies with our model may prioritize new approaches or actionable genes and pathways to eliminate therapy-resistant melanoma quiescent cells. This dual approach holds immense potential for melanoma treatment.

## Methods

### Reagents and tools table

| Reagent/resource | Reference or source | Identifier or catalog number |
|---|---|---|
| **Experimental models** | | |
| NOD.Cg-Prkdcscid Il2rgtm1Wjl/SzJ (NSG) | Charles River | N/A |
| **Recombinant DNA** | | |
| Tet-off H2B-GFP vector | Falkowska-Hansen et al, 2010 | |
| **Antibodies** | | |
| Mouse anti-human CD298-PE | BD | Cat.749741 (clone P-3E10) |
| Rat anti-human KI67-APC | Biolegend | Cat.151206 (clone 11F6) |
| Mouse anti-human GPNMB-PE | Invitrogen | Cat.12-9838-42 (clone HOST5DS) |
| Goat anti-human GPNMB | Biotechne | Cat.AF2550 |
| Anti-goat Alexa Fluor-647 | Thermofisher | Cat.A32849 |
| Anti-BrdU antibody | BD Bioscience | Cat. 347580 |
| **Oligonucleotides and other sequence-based reagents** | | |
| TTCATTGTGGGAGCAGAC | *RPLP0 Forward* | |

| Reagent/resource | Reference or source | Identifier or catalog number |
|---|---|---|
| CAGCAGTTTCTCCAGAGC | *RPLP0 Reverse* | |
| TCTGGCCCCTCAAACCTCACC | *ABCB5 Forward* | |
| TTTCATACCGCCACTGCCAACTC | *ABCB5 Reverse* | |
| CTCTTCTTGTTGCTGTGGG | *TYR Forward* | |
| GCTGAGTAGGTTAGGGTTTTC | *TYR Reverse* | |
| GCCTTGGAACTGGGACTGAG | *MITF Forward* | |
| CCGACGGCTGCTTGTTTTAG | *MITF Reverse* | |
| TTTCCAGCAATGAGAAACTC | *MMP2 Forward* | |
| GTATCTCCAGAATTTGTCTCC | *MMP2 Reverse* | |
| CGGTTCCGCCTGTCTCAAG | *MMP3 Forward* | |
| CGCCAAAAGTGCCTGTCTT | *MMP3 Reverse* | |
| CTTAGATCATTCCTCAGTGC | *MMP9 Forward* | |
| CGAGGACCATAGAGGTG | *MMP9 Reverse* | |
| AGGCAGGAGTAACAAGCTCAC | *EGFR Forward* | |
| ATGAGGACATAACAAGCCACC | *EGFR Reverse* | |
| CGAAACCATGAACTTTCTGC | *VEGF Forward* | |
| CCTCAGTGGGCACACACTCC | *VEGF Reverse* | |
| AACCTTCAACTCCTGCCTTCTCG | *AXL Forward* | |
| CAGCTTCTCCTTCAGCTCTTCAC | *AXL Reverse* | |
| GCTTCACACCTATTTCCCTGTC | *ITGB4 Forward* | |
| GACCCAGTCCTCGTCTTCTG | *ITGB4 Reverse* | |
| CCCAGCATCTGCAAAGCTC | *TGFB Forward* | |
| GTCAATGTACAGCTGCCGCA | *TGFB Reverse* | |
| GCAAGGTTCCTAGCGTGTAC | *VCAM Forward* | |
| GGCTCAAGCTGTCATATTCAC | *VCAM Reverse* | |
| GGGCATTTTTATGGCTTTCAAT | *STAT3 Forward* | |
| GTTAACCCAGGCACACAGACTTC | *STAT3 Reverse* | |
| AATGGGTCTGGCACCTACTG | *GPNMB Forward* | |
| GGCTTGTACGCCTTGTGTTT | *GPNMB Reverse* | |
| **Chemicals, enzymes and other reagents** | | |
| Iscove's modified Dulbecco's medium (IMDM) | Sigma-Aldrich-Merck | Cat. I3390 |
| L-glutamine | Euroclone | Cat. ECB3000D |
| Fetal Bovine Serum (FBS) | HyClone | Cat. SH30071.03IH |
| BrdU | Sigma-Aldrich | Cat. B5002 |
| Cytofix/Cytoperm Fixation/Permeabilization Kit | BD | Cat. 554714 |
| Collagenase type III | Worthington Biochem | Cat. LS004182 |
| Fibronectin | Roche | Cat. 11080938001 |
| Insert (8.0 μm pore size) | Corning | Cat. 353097 |
| 24-well plates | Corning-Falcon | Cat. 353097 |
| Crystal Violet solution (1%) | Sigma | Cat. V-5265 |
| Collagen Type I Rat Tail | Corning | Cat. 354249 |
| CyQUANT Direct Cell Proliferation Assay | Invitrogen | Cat. C35012 |
| Trametinib (GSK1120212) | MedChemExpress (MCE) | Cat. HY-10999 |
| Glembatumumab Vedotin | MedChemExpress (MCE) | Cat. HY-141604 |
| Poly-D-lysine | Life Technologies | Cat. A3890401 |
| Oligomycin | Merck Life Science | Cat. O4876 |
| FCCP | Santa Cruz | Cat. SC-203578 |
| Rotenone | Santa Cruz | Cat. SC-203242 |

| Reagent/resource | Reference or source | Identifier or catalog number |
|---|---|---|
| Antimycin A | Santa Cruz | Cat. SC-202467 |
| Matrigel Matrix HC | Corning | Cat. 354248 |
| L15 medium | Sigma-Aldrich | Cat. L5520 |
| Antigen Retrival Citra Solution | BioGenex | Cat. HK0686-9K |
| 4,6-Diamidino-2-phenylindole dihydrochloride (DAPI) | Sigma-Aldrich-Merck | Cat. 32670 |
| Quick-RNA MiniPrep Kit | Zymo Research | Cat. R1055 |
| OneScript Plus cDNA synthesis kit | Abm Industries | Cat. G236 |
| SYBR Green Master Mix | Applied Biosystem | Cat. 4385614 |
| Chromium Next 48 GEM Single Cell 3' Reagent kit (v3.1) | 10x Genomics | Cat. PN-1000075 |
| Live/Dead Fixable Viability Dye | BioLegend | Cat. 423103 |
| Riboblock RNAse Inhibitor | Life Technologies | Cat. EO0381 |
| RNeasy FFPE RNA extraction kit | Qiagen | Cat. 73504 |
| TruSeq RNA Library Preparation Kit (v2) | Illumina | Cat. RS-122-2002 |
| AMPure XP beads | Beckman Coulter | Cat. A63881 |
| **Software** | | |
| ImageJ Software | MD | RRID: SCR_003070 |
| ProCell (version 10.5.2 or higher) | BD | (Nobile, Bioinformatics, 2019) |
| FlowJo software (version 10.5.2 or higher) | BD | RRID: SCR_008520 |
| Fiji | | RRID: SCR_002285 |
| MATLAB software (version 2020a) | | RRID: SCR_001622 |
| QuPath software (v. 0.5.1) | | Bankhead, 2017 |
| Cell Ranger Single-Cell Software Suite (v6.0.1) | https://www.10xgenomics.com/support/software/cell-ranger/latest | |
| Seurat v4 package | | RRID: SCR_007322 (Stuart et al, 2019) |
| MsigDB website | http://software.broadinstitute.org/gsea/msigdb | |
| Cerebro App package | | Hillje et al, 2020 |
| *dpFeature* method in Monocle (v2.18) | | Trapnell et al, 2014 |
| HTseq | | Louphrasitthiphol et al, 2019 |
| DESeq bioconductor packages | RRID: SCR_000154 | Anders et al, 2015 |
| DAVID tool (version 6.8 Beta) | RRID: SCR_001881 | Huang et al, 2007 |
| Ingenuity Pathway Analysis (IPA) | | RRID:SCR_008653 |
| GSEA (Gene Set Enrichment Analysis) software v2.2.0. | | Subramanian et al, 2005 |
| **Other** | | |
| FACS Melody Cell Sorters | BD Bioscience | |
| FACS Aria | BD Bioscience | |
| PHERAstar FSX Microplate Reader | EuroClone | |
| Eclipse Ti2 spinning disc microscope | Nikon Instruments | |
| Thunder imaging system | Leica microsystems | |
| NanoZoomer S60 Digital Scanner | Hamamatsu Nanozoomer | C13210-01 |
| C1000 touch Thermo cycler | Biorad | |
| NovaSeq 6000 platform | Ilumina | |

## Patient-derived xenografts (PDXs) generation, characterization and in vitro culture

Tissue biopsies of metastatic melanomas were collected from patients whose informed consent was obtained in writing according to the policies of the Ethics Committee of the European Institute of Oncology and regulations of the Italian Ministry of Health. The studies were conducted in full compliance with the Declaration of Helsinki.

Metastatic melanoma PDXs were generated in NOD.Cg-Prkdcscid Il2rgtm1Wjl/SzJ (NSG) mice as previously reported (Bossi et al, 2016). For the herein described experiments, MM13 (NRAS mutated) and MM27 (BRAF mutated) PDXs were used for ex vivo and in vivo studies. For the in vitro functional experiments PDXs were maintained in culture in Iscove's modified Dulbecco's medium (IMDM, Sigma-Aldrich-Merck, Cat# I3390) supplemented with 200 mmol/L L-glutamine (Euroclone, Cat# ECB3000D) and 10% FBS (HyClone products Cytiva, Cat#SH30071.03IH).

## In vivo generation of H2B-GFP model

In vivo studies were approved and performed in our fully authorized animal facility and notification to the Ministry of Health (as required by the Italian Law; No 564/2019-PR), and in accordance with EU directive 2010/63. Mice of both genders were used at the age of 7–8 weeks.

MM13 and MM27 cells were transduced with the H2B-GFP reporter system (Falkowska-Hansen et al, 2010), which allows the tracking of cell division based on the inducible (Tet-off) expression system. In the absence of doxycycline, H2B-GFP is stably expressed; once doxycycline is added, the GFP signal is reduced at each cell cycle division, allowing the distinction of slow cycling (GFP-positive) and proliferating (GFP-negative) cell populations. In all, $2 \times 10^5$ PDX melanoma cells, stably infected with the H2B-GFP vector, were injected intradermally in the back of immuno-compromised mice. At palpability ($\cong$2 weeks), mice were treated with doxycycline via modified food pellets at a dose of 625 mg/kg for $\cong$3 weeks before sacrifice or, for the metastatic model, before tumor was resected at the volume of ~0.3 cm$^3$. Tumor volumes were monitored and annotated at least twice a week. Tumor volume was calculated using the modified ellipsoid formula: 1/2 (length × width$^2$). To monitor metastasis onset, mice were kept under doxycycline treatment for additional 3 weeks before sacrifice. Autopsies were performed to check for the presence of metastases.

## Fluorescence-activated cell sorting

Sorting was performed using fluorescence-associated cell sorters (Aria and Melody, BD Bioscience) to isolate GFP+ and GFP-populations from MM13 and MM27 PDXs after DAPI staining for live cells and from mice metastatic organs after labeling with human CD298-PE (BD, Cat#749741, clone P-3E10) and DAPI for human melanoma and live cells. Analysis was performed using the FlowJo software (BD; version 10.5.2 or higher).

## Modeling and simulation of cell proliferation

In the absence of doxycycline, the H2B-GFP fusion protein is continuously expressed in the nucleus and successfully incorporated in nucleosomes. During the chasing period, doxycycline in vivo administration shuts off H2B-GFP expression and proliferating cells lose half of their fluorescence signal upon each cell division. This allows tracking up to 8 divisions, after which point the H2B-GFP signal becomes indistinguishable from baseline cell autofluorescence (Falkowska-Hansen et al, 2010). In order to investigate the existence of multiple cell phenotypes in our samples, we relied on an in silico approach based on modeling and simulation. Specifically, we exploited ProCell (Nobile et al, 2019), a computational tool for cell proliferation analysis. ProCell assumes that cells divide asynchronously within a tumor and automatically calibrates a given mathematical model against label-retaining flow cytometry data. Specifically, it finds the optimal proportion of cells with distinct cell cycle entry properties —along with their specific division time intervals—such that the distance between the experimental fluorescence patterns and the fluorescence predicted by a computational simulation is minimized. An initial H2B-GFP fluorescence histogram, prior to doxycycline exposure, is required as input, to define the starting point of the simulation, as well as a target histogram obtained at any given time within the chasing period. Both histograms are truncated in order to exclude fluorescence values below the experimentally defined autofluorescence threshold. The optimization is automatically performed with Fuzzy Self-Tuning Particle Swarm Optimization, an efficient and settings-free swarm intelligence meta-heuristic (Nobile et al, 2019). In this work, the target fluorescence histograms were collected after 7 days. The search space (Table 1) was adapted to the specific experimental data as previously described (Nobile et al, 2019). Once the optimal parameters (cell proportions with corresponding means and standard deviations of division intervals) are identified, simulations of any length can be performed, in order to further validate the model against additional data. In our tests, we performed validation at 21 days. The fitness quantify the quality of the fitting of a model's simulation with respect to the experimental measurement (fluorescence histograms). The Hellinger distance provides a (positive) real value, where the lower the fitness, the better the model.

## In vitro bromodeoxyuridine (BrdU) incorporation

For in vitro short pulse chase, 3000 freshly sorted MM13 and MM27 GFP+ and GFP− cells were seeded in 96-well plates in triplicates and 10 µM of BrdU (Sigma-Aldrich, Cat# B5002) was added to complete culture medium. After 24 h cells were fixed in 70% of ethanol, denatured with 2 M HCl for 25′ and the reaction stopped adding Sodium Borate pH 8.5 for 2'. Cells were then stained with primary anti-BrdU antibody (BD Bioscience, Cat# 347580) and secondary antibody and analyzed by confocal microscopy (10X objective). All wells were acquired and GFP+ and BrdU+ cells were counted using ImageJ Software (National Institutes of Health, Bethesda, MD; RRID:SCR_003070).

## Immunofluorescence staining for FACS

For KI67 studies, freshly sorted GFP+ and GFP– MM13 and MM27 PDX melanoma cells were fixed and permeabilized with Cytofix/Cytoperm and Cytoperm Permeabilization Plus buffers (BD). Cells were then stained with human KI67-APC antibody (Biolegend, Cat#151206, clone 11F6).

**Table 1. Search space boundaries for mean cell division intervals for each population in each model.**

| Model | Quiescent cells | Proliferating cells | Fast proliferating cells | Slow proliferating cells |
|---|---|---|---|---|
| 1 | – | 21–504 h (0.01–40 h) | – | – |
| 2 | »504 h | 21–504 h (0.01–40 h) | – | – |
| 3 | » 504 h | – | 21–63 h (0.01–30 h) | 63–504 h (0.01–40 h) |
| 4 | – | – | 21–63 h (0.01–30 h) | 63–504 h (0.01–40 h) |

Corresponding standard deviation boundaries are shown in brackets. Cell proportion boundaries are by definition (0–1) in all cases.
Across all tests we performed the calibration using the following settings: swarm size of 50 particles; optimization length: 100 iterations; the statistics are based on 30 repetitions of each optimization.

For metastases studies, organs were dissociated mechanically and enzymatically with collagenase type III (Worthington Biochem, Cat# LS004182) at 37 °C to single cell suspension and GFP+ and GFP- cells were collected from each organ (lung, liver, kidney and lymph-nodes) and stained for human CD298 (PE conjugated) to mark human metastatic cells within mouse organs. Stained cells were analyzed by FACS (Celesta, BD Bioscience) and data were analyzed using the FlowJo software (BD; version 10.5.2 or higher; RRID:SCR_008520). For GPNMB and KI67 analysis, MM13 PDX melanoma cells were stained with human GPNMB-PE antibody (Invitrogen, Cat# 12-9838-42, clone HOST5DS), then cells were fixed and permeabilized as previously described for KI67 studies.

### Transwell migration assay

The migration assay was performed using 8.0 μm pore size (Corning, Cat#353093), fibronectin-coated (5 μg/cm² coating the outer part of the membrane, Roche, cat# 11080938001) inserts in 24-well plates (Corning-Falcon Cat# 353097). Triplicates of $1.5 \times 10^4$ freshly sorted MM13 and MM27 GFP+ and GFP- cells were plated in the upper chamber in IMDM serum-free medium. Complete medium was added to the lower chamber. After 24 h in a 37 °C incubator (20% $O_2$ for normal oxygen condition and 3% $O_2$ for low oxygen condition), non-migrated cells were discarded, while cells migrated to the lower surface of the inserts were stained with 0.5% Crystal Violet solution (50% Crystal Violet 1% Sigma Cat# V-5265, 35% ethanol in water). Four images of each insert were acquired at EVOS microscope and analyzed with the ImageJ Software to estimate the occupied area and calculate migration rate.

### Spheroid collagen invasion assay

In total, 1000 freshly sorted GFP+ and GFP- or GPNMB+ and GPNMB- MM13 and MM27 cells were grown as hanging drops in IMDM complete medium (Sigma-Aldrich-Merck, Cat# I3390) for 72 h at 37 °C in normal (20% $O_2$) and low (3% $O_2$) oxygen conditions. Single spheroids were harvested, resuspended in 1.5 mg/mL Collagen Type I Rat Tail (Corning, Cat# 354249) and plated on a 96-well plate. After 1 h at 37 °C at either 20%$O_2$ or 3% $O_2$, the solidified collagen gel was covered with IMDM supplemented with 2% FBS. The ability of cells to invade the area was monitored and images of 10 spheroids per group were collected at 24 h and 48 h using a 4× objective lens. The invasion area was analyzed with the ImageJ Software and normalized against the area at time 0.

### In vitro drug treatment viability assay

In vitro drug sensitivity was assessed by CyQuant (Invitrogen, Cat# C35012), plating 3000 freshly sorted GFP+ and GFP- or GPNMB+ and GPNMB- MM13 and MM27 cells in 96-well plates in triplicates. Cells were treated by a single exposure to either vehicle (DMSO), 10 nM trametinib (GSK1120212, catalog number A-1258) for 72 h at 37 °C at 20%$O_2$ or 3%$O_2$ or Glembatumumab Vedotin (MedChemExpress, Cat# HY-141604) (1ug/ml for GPNMB+ and GPNMB−) for 72 h at 37 °C in 20% $O_2$. After adding CyQuant reagent, fluorescence signal was acquired with PHERAstar FSX Microplate Reader (EuroClone, BMG Labtech) and the relative viability (%) was calculated upon normalization to DMSO controls.

### In vivo glembatumumab vedotin (GV) experiment

$2 \times 10^5$ MM13 cells were injected intradermally in immunocompromised mice (16 mice). When tumors were palpable, mice were randomized and treated iv with 5 mg/kg of GV once a week for 3 weeks ($n = 8$) or with vehicle as a control ($n = 8$). Tumor growth was monitored every 3 days using a digital caliper and resected when the tumor reached ~0.3 cm³ in volume. The statistical difference in tumor volume among the two groups was assessed by Student's *t* test. Survival analysis was calculated with GraphPad Prism 9.0, and differences among groups were estimated by using Log rank test. All mice were sacrificed when lymph-nodes were detected, autopsies were performed and all the organs were collected and checked for metastases.

### Assessment of mitochondrial function

Seahorse experiments were performed using XF Cell Mito Stress Kit (Seahorse Bioscience). Freshly sorted MM13 and MM27 GFP+ and GFP− cells were seeded at equal density of 40,000 cells into XF96 tissue microculture plates pre-coated with 50 μg/ml poly-D-lysine. Cells were incubated for 24 h in standard growth medium IMDM in a humidified incubator at 37 °C with 5% $CO_2$. After 24 h, the standard medium was replaced with XF DMEM Medium pH 7.4 supplemented with 25 mM glucose, 2 mM L-glutamine, and 1 mM sodium pyruvate. The cells were then incubated for 1 h at 37 °C without $CO_2$ before running the assay. The drug injection ports of the XF96 Assay Cartridge were loaded with assay reagents to a final concentration of 1 μM oligomycin (ATP synthase inhibitor, Merck Life Science, Cat# O4876), 2 μM FCCP (uncoupler, Dba Italia, Cat# SC-203578), 0.5 μM rotenone (Dba Italia, Cat# SC-203242), and 0.5 μM antimycin A (complex I/II inhibitors,

Dba Italia, Cat#SC-202467). OCR was measured under basal conditions and prior to and following additions of the metabolic modulators. Data were normalized to urea lysate protein levels and analyzed using Wave Desktop Software (Seahorse Bioscience), following manufacturer's instructions. For mitochondrial studies, GFP+ and GFP- cells were incubated with 50 nM MitoTracker Deep Red FM (mitochondrial mass) (Life Technologies, Cat#M22426) and 25 nM MitoTracker Orange CMTMRos (mitochondrial activity) (Life Technologies, Cat# M7510) for 30 min at 37 °C before cytometry analysis. All FACS data were acquired using a FACS (Celesta, BD Bioscience) and analyzed with FlowJo software.

### In vivo functional experiments

For in vivo functional studies, $10^4$ MM27 and MM13 GFP+ and GFP− cells were intradermally injected into the flank of NSG mice (5 mice per group) with Matrigel Matrix HC (Corning, Cat#354248) and L15 medium (Sigma-Aldrich, Cat#L5520). Tumor growth was monitored every 3 days using a digital caliper and resected when the tumor reached ~0.3 cm³ in volume. The statistical difference in tumor volume among the two groups was assessed by Student's $t$ test. Survival analysis was calculated with GraphPad Prism 9.0, and differences among groups were estimated by using Log rank test. All mice were sacrificed when lymph-nodes were detected, autopsies were performed and all the organs were collected and checked for metastases.

### Culture plates for controlled oxygen conditions

The culture plates for the experiments in controlled oxygen conditions have been detailed elsewhere (Becconi M et al) (Becconi M et al, submitted). For invasion assay, cells suspended in the collagen mixture were deposited on the surface of the well for the oxygen control. The collagen matrix and cell dispersion were prepared as detailed previously in the "spheroid collagen invasion assay paragraph", with cell quantities reported in the next paragraph.

### Timelapses of PDX melanoma cells under controlled oxygen condition

Migration and invasion assays for sorted MM13 and MM27 GFP+ and GFP− cells in the presence of controlled oxygen conditions were obtained with Nikon Eclipse Ti₂ spinning disc confocal89 and Leica Thunder microscopes. Migration and invasion experiments were performed for 16 and 8 h under controlled atmosphere and temperature (5% $CO_2$ and 37 °C). For migrations, after sorting, GFP+ and GFP− fractions were counted and 3000 cells seeded in 300 μl of growing medium in the modified well culture plates. Samples for invasion assays were prepared by seeding 6500 cells in 37 μl of 2.5 mg/ml collagen in growing medium. 300 μl of growing medium are then added in each well. Green (excitation 480/30 nm; emission 535/45 nm) fluorescence and differential interference contrast (DIC) images were collected every 20 min for 8 h for invasion assays and every 15 min for 16 h for migration assays; data from three independent samples were measured under controlled oxygen conditions for both GFP+ and GFP− cells.

### Quantitative analysis of trajectories of PDX melanoma cells under controlled oxygen conditions

Quantitative analysis of cell trajectories extracted from migration and invasion assays in the presence of controlled oxygen conditions were performed with scripts and computer routines written for Fiji (RRID:SCR_002285) (Schindelin et al, 2009) and MATLAB version 2020a software (RRID:SCR_001622). Segmentation was performed for both migration and invasion assays respectively on DIC and green fluorescent images by using the pixel classification method in Ilastik (Berg et al, 2019); training of the machine learning algorithm was performed on a single green fluorescent image for each timelapse, after applying on the original images a gaussian smooth filters available on Ilastik. Watershed algorithm implemented in Fiji was applied to refine identification of neighboring cells. Refinement of binarized labeled images were obtained employing an in-house developed MATLAB routine. Bidimensional and tridimensional tracking for migration and invasion assays were performed with TrackMate plug-in available in Fiji. Trackmate identification of single cells was performed by setting estimated blob diameter and threshold for the LoG detector equal to 10 and 0 μm, respectively; trajectories reconstruction was obtained with linking max distance and gap-closing max distance equal to 10 and 0 μm, respectively.

Cell position with respect to pixel coordinate in the bidimensional images of timelapse data were converted to displacement, *disp*, values as described in Fig. 4 and its caption. All the statistical analyses which followed cell trajectory reconstruction were performed with in-house developed MATLAB routines.

### Immunofluorescence on paraffin PDX sections and quantitative analysis

MM13 PDX primary tumors ($n = 2$) were collected, formalin-fixed and paraffin-embedded. Sections were processed for deparaffinization and antigen retrieval (10 mM citrate buffer), blocked in 4% BSA + 0.05% Tween20 in TBS 1× and incubated overnight with anti-human GPNMB (Biotechne catalog #AF2550), followed by secondary antibody staining (Thermofisher anti-goat Alexa Fluor-647, cat. A32849). Samples were then washed (TBS 1× + 0.05% Tween20) and fixed in paraformaldehyde 4%. Slides were counterstained with DAPI (Sigma-Aldrich-Merck, cat. 32670-25MG-F) for nuclei labeling and mounted with Mowiol and coverslip.

A sequential slice of each sample was stained with Hematoxylin Eosin (H/E) for the spatial annotation of hypoxic/ peri-hypoxic regions, performed by an experienced pathologist based on histopathological criteria. Whole slide images were acquired with NanoZoomer S60 Digital slide scanner (C13210-01), equipped with 20×/0.75NA and analysed with QuPath software (v. 0.5.1) (Bankhead et al, 2017). For fluorescence analysis, nuclei were segmented with a pre-trained Stardist model (Schmidt et al, 2018) and the cytoplasmic area was represented as a band of thickness 2 μm around the nucleus, using the related QuPath function. GPNMB expression was analysed by creating an object classifier for GPNMB mean intensity in the cell, and then data were plotted by comparing GPNMB-positive cells percentage in normoxic and peri-hypoxic areas of each samples.

## Quantitative RT-PCR

MM13 GPNMB+ and GPNMB- total cellular RNA was isolated with the Quick-RNA MiniPrep (Zymo Research, Irvine, CA; Cat# R1055), following the manufacturer's instructions. RNA was reverse transcribed into cDNA using OneScript Plus cDNA synthesis kit (Abm Industries, San Francisco, CA; Cat# G236), following the manufacturer's instructions. Real-time PCR was performed on a Biorad C1000 touch Thermo cycler using SYBR Green Master mix (Applied Biosystem, Waltham, MA, Cat# 4385614) and gene-specific primers (see "Reagents and Tools Table").

## Sample preparation for scRNA-seq

Three primary MM13 PDXs and four matched lymph nodes (1 for primary tumor#1 and #2 and 2 for primary tumor#3) freshly sorted for GFP expression were counted, resuspended in an appropriate volume of 0.04% BSA/PBS (10x Genomics guidelines) and submitted for sequencing. On average, 2000–3000 cells per sample were used for sequencing. Cells were processed and libraries were prepared with Chromium Single Cell 3' Reagent kit v2 (10x Genomics), Chromium Single Cell 3' Reagent kit v3 (10x Genomics), or Chromium Next 48 GEM Single Cell 3' Reagent kit v3.1 (10x Genomics). Paired-end sequencing at high coverage (50,000 reads per cell) was performed on Illumina NovaSeq 6000 platform.

## Single-cell RNA-seq data pre-processing and analysis

Sequencing results were converted to FASTQ format using Illumina bcl2fastq software. The Cell Ranger Single-Cell Software Suite v6.0.1 (https://www.10xgenomics.com/support/software/cell-ranger/latest) was used to perform barcode processing and single-cell 3' gene counting. The cDNA insert was aligned to the GRCh38 human reference genome. Only confidently mapped, non-PCR duplicates with valid barcodes and UMIs were used to generate the gene-barcode matrix containing 15,627 cells. Further analysis, including quality filtering, identification of highly variable genes, dimensionality reduction, standard unsupervised clustering algorithms, and discovery of DEGs, was performed using the Seurat v4 package (RRID:SCR_007322) (Stuart et al, 2019). Additional conservative cut-offs were further applied based on the number of genes detected per cell (>100), and the percentage of mitochondrial unique molecular identifier (UMI) counts (< 30). To limit background noise and increase robustness of differential analysis only cells with at least 4000 features were selected for further analysis (14,853). Data integration was performed using Seurat v4 package using canonical correlation analysis (CCA) with FindIntegrationAnchors function and final integration across conditions with IntegrateData. PCA analysis was performed on integrated dataset to reduce data dimension and the cells were clustered in a two-step process. First, the cells were embedded in a k-nearest neighbor (KNN) graph and then, the Resolution-Optimized Louvain algorithm was applied. The dimensionality reduction and clustering visualization was performed using UMAP (Uniform Manifold Approximation and Projection) approach. The marker genes that define the clusters were then found via Seurat's

FindAllMarkers function using the 'wilcox' method (Wilcoxon rank sum test).

The marker genes that define the QQ in primary tumor and the QQ in metastases were found via Seurat's FindAllMarkers function versus all the other populations. Venn Diagram was used to highlight common genes. Cell cycle score assignment was performed using CellCycleScoring function from Seurat package. Gene set variation analysis (GSVA R package with RNA-seq mode) was used to identify the enrichment of each gene set across clusters and cell populations (Hänzelmann et al, 2013). False discovery rate adjusted $P$ values (qFDR) were computed using the method of Benjamini and Hochberg. "GO biological processes", "KEGG Canonical pathways" and the "Hallmark gene sets" were obtained from the MsigDB website (http://software.broadinstitute.org/gsea/msigdb) as well the 'growth inhibitory genes' gene set (https://www.gsea-msigdb.org/gsea/msigdb/cards/module_488) and the cell cycle-related gene set. Gene sets with FDR adjusted $P$ value qFDR <0.05 were considered significant. Cerebro App package (Hillje et al, 2020) was used for interactive visualization of single cell RNA-seq data. The dataset containing FACS sorted GFP-positive and negative populations was grouped in four classes (GFP + /KI67− (QQ), GFP + /KI67 + (QP), GFP-/KI67−(PQ) and GFP−/KI67+ (PP) based on median KI67 expression score. Re-clustering of QQ and PQ cell states was performed by PCA analysis and subsequent KNN-graph and Louvain alogorithm based clustering based on expression 2000 most variable features. Trajectory and pseudo-temporal gene analysis were performed using Monocle 2.20.0 version. Single-cell pseudotime trajectories were constructed using dpFeature method in Monocle v2.18 (Trapnell et al, 2014).

## RNA-seq bulk on KI67 positive and negative fixed melanoma PDX

MM13 PDXs cells were incubated with LIVE/DEAD Fixable Viability Dye (BioLegend, Cat# 423103), then fixed using PFA 1% supplemented with Riboblock RNAse Inhibitor (Life Technologies, Cat# EO0381), permeabilized with saponin 0.1% supplemented with RNAse Inhibitor and stained for KI67-647 antibody. Samples were sorted at 4 C° for KI67 expression and subpopulations collected. RNA extraction was immediately performed using the RNeasy FFPE RNA extraction kit (Qiagen, Cat# 73504) according to manufacturer's instructions.

RNA-seq libraries were prepared from 1 μg of total RNA using the TruSeq RNA Library Preparation Kit v2 (Illumina, Cat#RS-122-2002) according to manufacturer's instructions. The libraries were additionally purified using AMPure XP beads (Beckman Coulter, Cat# A63881), quality checked at Agilent 2100 Bioanalyzer and 50 bp paired-end sequenced on an HiSeq 2000 Sequencing System (Illumina). RNA-seq reads were aligned to genome (GRCh38) using TopHat2 2.0.9 (Kim et al, 2013) starting from $3 \times 10^7$ mapped paired-end reads per sample. Read counts of each gene were quantified using HTseq (92) and differential analysis was performed using DESeq bioconductor packages (RRID:SCR_000154) (Anders et al, 2015). Genes were identified as differentially expressed (DEGs) when the following criteria were met: log2 fold change (FC) ≥ |1|, $P$ value < 0.05. DEGs were analyzed with Gene Ontology term enrichment using DAVID tool (version 6.8 Beta) (RRID:SCR_001881) (Huang et al, 2007),

Ingenuity Pathway Analysis (IPA) (RRID:SCR_008653) and with GSEA (Gene Set Enrichment Analysis) software v2.2.0 (Subramanian et al, 2005). Gene sets enriched at FDR < = 0.05 or 0.1 were considered statistically significant.

## Quantification and statistical analysis

Data are represented as mean ± SD of biological triplicates (if not, differently indicated in the text). Comparisons between two or more groups were assessed by using unpaired two-tailed Student's *t* test. For the statistical difference in tumor volume among different populations, unpaired *t* test was used. For all the statistical tests: ns, not significant; $*P < 0.05$; $**P < 0.01$; $***P < 0.001$; $****P < 0.0001$.

## Data availability

Melanoma single cell RNA-seq and bulk RNA-seq data have been deposited at Gene Expression Omnibus (GEO) with accession number GSE288622.

The source data of this paper are collected in the following database record: biostudies:S-SCDT-10_1038-S44319-025-00501-w.

## Peer review information

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

## Acknowledgements

This work was supported by the Ricerca Finalizzata 2018 Grant GR-2018-12367747 to FL, AIRC grant IG 2023 ID 28890 and the Italian Ministry of Health with Ricerca Corrente and 5×1000 funds. The authors wish to thank Alberto Gobbi and Manuela Capillo (Cogentech) for the excellent support in animal work, Serena Stinà, the IEO Genomic, Flow Cytometry and Imaging Units for the considerable technical support. FM is supported by Fondazione Umberto Veronesi.

## Author contributions

**Fiorenza Lotti**: Conceptualization; Resources; Data curation; Formal analysis; Funding acquisition; Validation; Investigation; Visualization; Methodology; Writing—original draft; Project administration; Writing—review and editing. **Marine Melixetian**: Data curation; Software; Formal analysis; Investigation; Visualization; Methodology; Writing—review and editing. **Thalia Vlachou**: Formal analysis; Visualization; Methodology; Writing—review and editing. **Marco S Nobile**: Resources; Software; Formal analysis; Visualization; Writing—review and editing. **Leone Bacciu**: Software; Formal analysis; Visualization; Methodology. **Marco Malferrari**: Data curation; Software; Formal analysis; Investigation; Visualization; Methodology; Writing—review and editing. **Nicolò Quaresima**: Data curation; Software; Formal analysis; Visualization; Methodology. **Stefania Rapino**: Resources; Validation; Visualization; Writing—review and editing. **Federica Marocchi**: Visualization; Methodology; Writing—review and editing. **Massimo Barberis**: Methodology; Writing—review and editing. **Chiara Soriani**: Methodology. **Barbara Gallo**: Methodology. **Velia Mollo**: Methodology. **Ilaria Ferrarotto**: Methodology. **Daniela Bossi**: Methodology. **Pier Francesco Ferrucci**: Investigation; Writing—review and editing. **Pier Giuseppe Pelicci**: Conceptualization; Writing—original draft; Writing—review and editing. **Lucilla Luzi**: Data curation; Software; Formal analysis; Validation; Investigation; Writing—review and editing. **Luisa Lanfrancone**: Conceptualization; Resources; Supervision; Funding acquisition; Validation; Writing—original draft; Project administration; Writing—review and editing.

Source data underlying figure panels in this paper may have individual authorship assigned. Where available, figure panel/source data authorship is listed in the following database record: biostudies:S-SCDT-10_1038-S44319-025-00501-w.

## Disclosure and competing interests statement

The authors declare no competing interests.

