## [Peer Review File · EMBO Reports]

GPNMB marks quiescent cell populations in melanoma and promotes metastasis formation

Luisa Lanfrancone, Fiorenza Lotti, Marine Melixetian, Thalia Vlachou, Marco S. Nobile, Leone Bacciu, Marco Malferrari, Nicolo' Quaresima, Stefania Rapino, Federica Marocchi, Massimo Barberis, Chiara Soriani, Barbara Gallo, Velia Mollo, Ilaria Ferrarotto, Daniela Bossi, Pier Francesco Ferrucci, Pier Giuseppe Pelicci, and Lucilla Luzi

Corresponding author(s): Luisa Lanfrancone (luisa.lanfrancone@ieo.it)

Review Timeline:

Submission Date:	11th Dec 24
Editorial Decision:	24th Jan 25
Revision Received:	15th Apr 25
Editorial Decision:	13th May 25
Revision Received:	26th May 25
Accepted:	2nd Jun 25

Editor: Achim Breiling

Transaction Report:

Dear Dr. Lanfrancone

Thank you for the transfer of your manuscript to EMBO reports. I have now received the reports from the three referees that were asked to evaluate your study, which can be found at the end of this email.

As you will see, the referees think that these findings are of high interest. However, they have several comments, concerns, and suggestions, indicating that a major revision of the manuscript is necessary to allow publication of the study in EMBO reports. As the reports are below, and all the referee concerns need to be addressed, I will not detail them here.

Given the constructive referee comments, I would like to invite you to revise your manuscript with the understanding that the concerns of the referees must be addressed in the revised manuscript and in a detailed point-by-point response. Acceptance of your manuscript will depend on a positive outcome of a second round of review. It is EMBO reports policy to allow a single round of revision only and acceptance of the manuscript will therefore depend on the completeness of your responses included in the next, final version of the manuscript.

Revised manuscripts should be submitted within three months of a request for revision. Please contact me to discuss the revision (also by a video call) if you have questions or comments regarding the revision, or should you need additional time.

- 1) a .docx formatted version of the final manuscript text (including legends for main figures, EV figures and tables), but without the figures included. Figure legends should be compiled at the end of the manuscript text.
- 2) individual production quality figure files as .eps, .tif, .jpg (one file per figure), of main figures and EV figures. Please upload these as separate, individual files upon re-submission.

- 4) a complete author checklist, which you can download from our author guidelines (<https://www.embopress.org/page/journal/14693178/authorguide>). Please insert page numbers in the checklist to indicate where the requested information can be found in the manuscript. The completed author checklist will also be part of the RPF.

- 5) that primary datasets produced in this study (e.g. RNA-seq, ChIP-seq, structural and array data) are deposited in an

appropriate public database. If no primary datasets have been deposited, please also state this in a dedicated section (e.g. 'No primary datasets have been generated and deposited'), see below.

The accession numbers and database should be listed in a formal "Data Availability" section that follows the model below. This is now mandatory (like the COI statement). Please note that the Data Availability Section is restricted to new primary data that are part of this study. This section is mandatory. As indicated above, if no primary datasets have been deposited, please state this in this section

Data availability

8) Regarding data quantification and statistics, please make sure that the number "n" for how many independent experiments were performed, their nature (biological versus technical replicates), the bars and error bars (e.g. SEM, SD) and the test used to calculate p-values is indicated in the respective figure legends (also for EV and Appendix figures). Please also check that all the p-values are explained in the legend, and that these fit to those shown in the figure. Please provide statistical testing where applicable. Please avoid the phrase 'independent experiment', but clearly state if these were biological or technical replicates. Please also indicate (e.g. with n.s.) if testing was performed, but the differences are not significant. In case n=2, please show the data as separate datapoints without error bars and statistics. See also: <http://www.embopress.org/page/journal/14693178/authorguide#statisticalanalysis>

9) Please add scale bars of similar style and thickness to microscopic images, using clearly visible black or white bars (depending on the background). Please place these in the lower right corner of the images themselves. Please do not write on or near the bars in the image but define the size in the respective figure legend.

10) Please also note our reference format:

12) We now use CRedit to specify the contributions of each author in the journal submission system. CRedit replaces the author contribution section. Please use the free text box to provide more detailed descriptions and do NOT provide your final manuscript text file with an author contributions section. See also our guide to authors: <https://www.embopress.org/page/journal/14693178/authorguide#authorshipguidelines>

13) All Materials and Methods need to be described in the main text using our 'Structured Methods' format, which is required for

all research articles. According to this format, the Methods section should include a Reagents and Tools Table (listing key reagents, experimental models, software, and relevant equipment and including their sources and relevant identifiers), uploaded as separate file, and a Methods section in which we encourage the authors to describe their methods using a step-by-step protocol format with bullet points, to facilitate the adoption of the methodologies across labs. More information on how to adhere to this format as well as downloadable templates (.doc) for the Reagents and Tools Table can be found in our author guidelines (section 'Structured Methods'):

14) Please add up to 5 keywords to the manuscript and order the sections like this, using these names:

Title page - Abstract - Keywords - Introduction - Results - Discussion - Methods - Data availability section - Acknowledgements - Disclosure and Competing Interests Statement - References - Figure legends - Expanded View Figure legends

15) Please make sure that all the funding information is also entered into the online submission system and that it is complete and similar to the one in the acknowledgement section of the manuscript text file.

I look forward to seeing a revised form of your manuscript when it is ready.

Yours sincerely,

Referee #1:

In their manuscript entitled "Challenging the quiescent paradigm in melanoma: GPNMB drives metastasis formation", Fiorenza and colleagues describe the identification and analysis of a quiescent melanoma subset using intricate in vitro and in vivo studies with patient-derived melanoma cell lines. Using an elegant and carefully conducted combination of in vivo label retention and the measurement of Ki67 and BrdU incorporation, they identify in two independent models a proportion of quiescent melanoma cells that is surprisingly high, but that goes along with mathematical modeling. They find that both GFP+ (label-retaining and "non-dividing") as well as GFP- (label-losing and dividing) cells are able to occur in Ki67 positive and -negative states, indicating high heterogeneity and the transient occurrence of cell states. In subsequent single cell analyses, the authors identify stress signatures characteristic for four different cell states they term quiescent QQ (GFP+ and Ki67-negative) and PQ (GFP- and Ki67-) as well as proliferative PP (GFP- and Ki67+) and QP (GFP+ and Ki67+).

Intriguingly, PQ and QQ cells contain partially overlapping stress and cell death signatures in contrast to the proliferative cells. In addition, they have transcriptional features associated with therapy resistance. In subsequent analyses, the authors focus more on the GFP+ cells and find an enhanced migratory and invasive behaviour that goes along with increased mitochondrial activity. In vivo, GFP+ cells show a higher propensity for metastasis, and the proportion of QQ cells is higher in metastases compared to primary melanoma. Finally, they use their single cell data and bulk sequencing of Ki67-negative melanoma cells to identify markers for metastatic quiescent cells. Using this, they identified the surface protein GPNMB as strongly enriched in this population. GPNMB positive melanoma cells have similar in vivo properties to GFP+ cells, and the authors successfully used Glematumumab vedotin (GV), a GPNMB-targeting cancer drug, to target metastasis development in the mouse melanoma models.

The study is highly interesting and was very carefully conducted and controlled, using two animal models and including publicly available datasets. It is particularly elegant that their findings come together in the application of a drug that is already used in clinical trials for several tumor entities including melanoma. Overall, the data are very convincing. The authors present a complex study encompassing a wealth of data. In my opinion, they address all major relevant research questions of this topic.

However, some questions arose that need to be addressed in order to better explain the methods or interpretation of the findings.

- 1) It would be useful to mention the presence of common melanoma driver and tumor suppressor genes of the melanoma cell lines in the Materials and Methods section.
- 2) Currently, the authors have only deposited their scRNA seq data under a private link (GSE251865), but they did not mention the deposition of the bulk RNA seq data. This should be amended, and a reviewer link for the datasets should be generated.
- 3) Figure 1B/Supplementary Figure 2A: Please explain shortly how the fitness (Hellinger distance) is interpreted. How would a perfect fit look like?
- 4) Figure 2D: The authors explain that "KI67 expression was used as a marker of active G1-S-G2M cell cycle phases, while BrdU incorporation was used as marker of DNA synthesis (24 hour pulse)."

However, Figure 2D shows a lack of KI67 expression in G1 cells. Can the authors comment on the question of sensitivity (how much of the cell cycle phase is covered by these treatments)? How likely is it to miss a signal in a cycling cell?

5) Text to Figure 2 and Supplementary Figure 3: "Analyses of several published melanoma signatures showed increased expression of invasion-related pathways ("MITF targets", "Invasion" and "MSC" melanoma signatures) in GFP+ cells, and of proliferation- and melanocyte-associated pathways ("Mitotic" and "Pigmentation" signatures) in GFP- cells (Fig. S3A)"

This is hard to see in the UMAP visualisation, as GFP+ and GFP- segregate minimally (as the authors state themselves). I would recommend to omit the UMAP visualisations and only refer to the violin plots.

6) Figure 2D Figure legend: "UMAP visualization of 14,353 MM13 melanoma cells analysed by scRNA-seq and integrated across 3 different melanoma primary tumors with 4 matched lymph node metastatic lesions..."

It only becomes clear much later in the paper (Figure 7) where the primary tumors and lymph node metastases are coming from. Please refer to this section of the paper to avoid confusion.

7) Supplementary Figure 4E: " Moreover, analyses of a recently published melanoma signatures from patient biopsies (52) showed that our QQ/PQ cell states are enriched in the expression of melanoma states associated to acquired resistance to immunotherapy (Fig. S4E)"

Please provide references describing the involvement of the signatures in acquired resistance to therapy. Also, which datasets were compared to the QQ data to calculate the indicated p values?

8) Figure 3C: "Conversely, State 3 was enriched in QP and PP cells"

According to the legend, showing states 1,2, and 3, it seems to me that this statement is be more relevant for state 2. However, the colour legend is tiny, and it is not easy to differentiate between green and blue (at least for me). Please check this statement and increase the size of the colour legend.

Shortly after this (page 5 of the Results section), the authors write: "...transiting from a quiescent (QQ) to an active proliferative state (PP and PQ), and back to a quiescent (PQ) state."

It seems that "(PP and PQ)" contains a typo that should be corrected to "(PP and QP)".

9) Fig. 4D; Fig. S5D: "Furthermore, comparison of the mitochondrial respiratory activities of freshly isolated MM13 and MM27 GFP+ and GFP- cells revealed increased mitochondrial respiration in GFP+ cells, with no change in mitochondrial mass".

What is the data basis for the latter statement ("no change in mitochondrial mass")? Please comment.

10) Fig. 5E-F; Fig S6D-E: "GFP+ populations showing improved migratory and invasive properties in hypoxic conditions in the first 8-16 hours of the experiments..."

This statement refers to the 2D experiments in Figure 5E and Figure S6D. In the 3D experiments, improved invasive properties are seen until 2-3 hours (Figure 5F, Figure S6E. Please correct this statement.

Also, the resolution of Figure 5E needs to be improved.

11) In Figure 6H and Figure S7G, the authors describe "larger metastases within organs, as evidenced by the increased number of human CD298+ cells quantified upon sacrifice (day 100 post injection)". In their figure legend, they define the metastasis size by "average number of organs with at least 50% of melanoma cells per mouse".

This is a lot of tumor load in the respective organs, and it is surprising that the mice survived this long without the need for prior sacrifice. Could the authors please comment on this and explain in better detail how the analysis of metastases was performed and how the numbers were calculated (in a three-dimensional organ context)?

12) Figure 7F, G, Figure S8D: "Cell populations with primary QQ cells enriched at early stages of the trajectory, while metastatic QQ cells enriched at later points of pseudotime (Fig. 7F-G). Interestingly, the quiescent PQ cells showed an opposite trend being less enriched in early stage in primary and more enriched at later stage in metastasis (Fig. S8D)."

As the general pseudotime course, shown in the respective density plots in Figure 7G and S8D is more or less similar between QQ and PQ cells, I find it counterintuitive to call this an "opposite trend".

13) Figure 8B: "GPNMB was highly expressed in QQ and PQ cells compared to other cell states (Fig. 8B)."

A reference to Figure 3A would be useful here.

14) Figure 8F, Figure S9D: "Indeed, the percentage of GPNMB+/KI67- cells rises to 91,3% in the lymph node metastases"

Do these figures show one example of a primary tumor and a corresponding lymph node metastasis, or are the data derived from pooled samples?

Typos:

1) Introduction, page 1: ",,... are though to play a crucial role..."

Should be: "... are thought to play a crucial role..."

2) Legend to Supplementary Figure 1C: "...show cell proliferation (gradual GFP lost) over time...)"

Should be(gradual GFP loss)...

3) Figure S9D: Typo in the labeling of the GPNMB+ LN met cells: lower section should be Ki67-

4) Last page of the Discussion: "Our research strengthens this possibility by identifying GPNMB as a specific marker for targeting the pro-metastatic and drug resistance phenotypes of quiescent melanoma cells."

Should be: "pro-metastatic"

Referee #2:

In the context of cancer, quiescence is thought to concern a small sub-subpopulation of cells within the tumors that is capable of reverting to a growth state and leads to both relapse and drug resistance. In this manuscript, Lotti et al challenge this concept in cutaneous melanoma by showing that quiescence is accessible to most, if not all, tumor cells. They also provide a comprehensive analysis of quiescent melanoma cells and identify a novel (GPNMB) marker of this cell state that can be targeted to mitigate pro-metastatic ability and development of drug resistance. Specifically, Glembatumumab vedotin, a GPNMB inhibitor, induces the death of quiescent melanoma cells.

This article contains a large amount of data. It well-written and experiments have been carefully designed. The model system is extremely well-described and has been used successfully in previous publications by the same group. In my opinion, the results presented here may also be of interest for understanding the process of quiescence, metastatic spread and drug resistance in other cancer models.

Below are my comments to be considered by the authors.

Results:

Page 9: the authors wrote "In melanoma, low oxygen (LO) conditions drive the ability of melanoma cells to switch between different states, to develop therapy resistance and to exhibit increased expression of markers associated with cancer stemness, particularly those related to slow-cycling properties".

I would suggest the authors to add the reference "Cheli et al" (Oncogene 2012, PMID: 21996743) which is highly related to this section.

Figure 8:

- Do oxygen concentrations influence GPNMB expression?
- Is forced expression of GPNMB in GPNMB-negative cells sufficient to render them more quiescent and affect their motility or MEKi sensitivity?

Figure 8F: The authors show that GPNMB expression is associated with both quiescent and metastatic phenotype, with the percentage of GPNMB+/KI67- cells rising to 91,3% in the lymph node metastases.

Is the percentage of GPNMB+/KI67- cells is also up-regulated in other metastatic sites (lung, liver...)? Can the authors comment on this?

Figure 8O: While GPNMB negative cells have a significantly shorter latency to resection (fig8L), the authors did not observe the same effect when inhibiting GPNMB with the ADC? Can the authors comment on this point?

Discussion:

Page 15: The authors show that primary-tumor QQ cells displayed enriched pathways related to iron ion homeostasis. Would a ferroptotic drug be of interest in the adjuvant setting? This point could be discussed in the discussion section.

Page 16: The authors suggest that the definition of the quiescent state and of the pathways activated/deactivated in these cells may vary between cell lines or patient-derived cells, and this definition still requires further characterization.

Related to this sentence, I would suggest the authors to discuss the manuscript from "Cheli et al. (Oncogene 2011 PMID: 21278797)". Cheli et al showed that (i) slow-cycling CFSE retaining cells have an increased capability of migrating in vitro, (ii) displayed an increased p27 expression, which is required for exacerbation of the tumorigenic properties of melanoma cells and (iii) increased expression of SPARC, but (iv) displayed low MITF expression.

Could it be that slow-cycling CFSE retaining cells (Cheli et al. PMID: 21278797)" and GFP+ cells (Lotti et al herein) share molecular features (enhanced motile ability) and markers (p27, SPARC) but represent different stages in the transition towards the quiescent state with also different markers (MITF, GPNMB)?

Minor:

Fig7C (right panel): I would suggest swapping KI67 expression in the primary tumor and metastases in the graph to follow the same order as in the other panels.

Figure 8: Could the authors please indicate on the figure panels and also in the figure legend which panels correspond to the QQ and PQ cells.

Figure 8N: Can the authors describe in the "NT" caption (I assume these are non treated cells).

Referee #3:

This manuscript uses a tagging approach to identify quiescent cells in xenograft tumours and functionally assess their metastatic ability. Essentially, they use a dox repressible system to turn off H2b-GFP expression and define quiescent cells as the GFP hi cells after some time. There is a lot of important detail missing from these experiments that makes it difficult to understand some parts of this manuscript. The potential shortcoming of this system is that it requires doxycycline to repress marker gene expression.

The models are then grown as xenografts to enhance the biological relevance of the models, but this may also introduce unwanted problems. We do not know if repression of the entire tumour can be maintained in vivo.

The authors identify apparently hypoxic regions within the tumour which is a common feature of xenografted models and regions of poor blood vessel infiltration. Thus, some of the GFP+ cells may still represent proliferating cells where derepression has occurred. Also as recognised by authors, the tumour is in dynamic change with quiescent cells re-entering the cell cycle and cycling cells exiting into quiescence.

After figure 1 where they show this dynamic effect, they go on to use better markers of quiescence such as BrdU incorporation and Ki67 expression but persist with the broad classification of GFP+ and GFP- representing quiescent and proliferating cells. I think that this leads to a lot of unnecessary confusion. The subsetting of cells in QQ, PP, QP, PQ for example does not really give any new insights and I think is flawed.

The expression analysis is performed at a single time point (I think that is the case, the details of this were lacking) so are a snapshot of cells at that time. Unless QP and PQ subset cells are caught just as they are transitioning from quiescence to proliferation or the reverse, it is difficult to understand how they can have significantly different transcriptional programs.

In Figure 3A, the UMAPs show again that GFP status is a very poor indicator of quiescence or proliferation. The Ki67- GFP-+ population overlap completely, as do the Ki67+ populations in Fig 3A. I think the data in Fig 3B is confounded by the GFP+ Ki67+ being derepression of GFP expression in proliferating cells, and cells that have lost GFP expression possibly and exited cell cycle.

The pathways identified in the PP and QP cells are almost identical, and overlap better than the QQ and PQ gene sets. If these different populations represented some particular spatial region of the tumour e.g. in the vicinity of hypoxic regions this might provide support for these dynamic populations representing functionally different forms.

The major conclusion from this study is that less proliferative/quiescent cells have an enhanced metastatic potential. This is very similar to the MITF low MITF high effect. Is this really any different?

Finally, the identification of GPNMB is interesting although the authors have not clearly stated their purpose here. Also, in Fig 8H, there is a subset of Ki67+ mets with very high GPNMB suggesting it is not the strongest candidate for a marker of quiescent cells. The other data generated show statistical significance but there is little quantitative difference observed.

The strongest data is the GPNMB directed ADC which clearly blocks metastasis and is strong evidence that GPNMB expressing cells are a metastatic population. Indeed, this is a very clear demonstration that quiescent cells are indeed more metastatic than their proliferating counterparts. If this was made more clearly the conclusion of this study it would be suitable for publication in this journal.

As it is at present, the authors present a very jumbled story that is very difficult to follow, and makes unnecessary claims not strongly backed by evidence.

Specific points:

Figure 1C-E; Figure 3; at what time point were these cells analysed? 3 weeks after Dox addition? This information must be in the figure legend. There is no indication what time after dox induced repression the scRNAseq data in figure 2 was generated? Are the cells isolated from PDX generated under the same conditions and time of exposure to dox as the scRNAseq experiments? If so, the GFP+/- subsetting is a flawed approach as indicated by the scRNAseq analysis.

Figure 4B; The reduced sensitivity is likely because the the GFP+ cells are not proliferating and thus likely to be less sensitive. Proliferating cells are more sensitive to MEKi.

Figure 4D; The mitotracker data indicates that there is more mitochondrial mass in the GFP+ cells. As the data in Fig 4D is from a double staining of MitoTr Orange that measures mitochondrial activity and MitoTr far red measuring mito mass, my guess is that the histogram is the far red and the bar graph is the ratio of orange and red signals. How this is not defined clearly in the legend. Why not show this data as a dot plot of the GFP- and GFP+ on the same graph of Red v orange fluorescence intensity. I cannot understand the data as it is presented currently.

Figure 4E; If the MitoTr red (PE) is showing more mitochondria then the increased O2 consumption is an expected result. The difference in mitochondrial mass is surprising.

Figure 5G; MITF hi, G1 phase cells are known to be faster migrating.

Figure 7A; Problem with the %GFP+ cells in lymph nodes in the much higher background fluorescence in that sample. This could significantly affect the % GFP+. A control of lymph node from non-tumour bearing mice would provide a better cut-off.

Figure 7B; Is it a surprise that primary and mets have different expression profiles? I would expect to see difference in both the Ki67- and Ki67+ population between primary and mets as mets are in a very different environment and therefore must change their transcriptome to allow them to survive in the new environment.

Figure 7c; A big difference here is the GFP+ primary and GFP- met have lost their cluster 0 from figure 2A, but cluster 0 is very prominent in the other population in each case.

Page 12, 2nd paragraph; I don't understand your conclusion? How is this promoting dormancy? This isn't a lot of difference between primary and mets in terms of Ki67- cells.

Figure 5E legend; is this 3 experiments or 3 cells?

Point-by-point response to reviewers

Referee #1:

In their manuscript entitled "Challenging the quiescent paradigm in melanoma: GPNMB drives metastasis formation", Fiorenza and colleagues describe the identification and analysis of a quiescent melanoma subset using intricate *in vitro* and *in vivo* studies with patient-derived melanoma cell lines. Using an elegant and carefully conducted combination of *in vivo* label retention and the measurement of Ki67 and BrdU incorporation, they identify in two independent models a proportion of quiescent melanoma cells that is surprisingly high, but that goes along with mathematical modeling. They find that both GFP+ (label-retaining and "non-dividing") as well as GFP- (label-losing and dividing) cells are able to occur in Ki67 positive and -negative states, indicating high heterogeneity and the transient occurrence of cell states. In subsequent single cell analyses, the authors identify stress signatures characteristic for four different cell states they term quiescent QQ (GFP+ and Ki67-negative) and PQ (GFP- and Ki67-) as well as proliferative PP (GFP- and Ki67+) and QP (GFP+ and Ki67+).

Intriguingly, PQ and QQ cells contain partially overlapping stress and cell death signatures in contrast to the proliferative cells. In addition, they have transcriptional features associated with therapy resistance. In subsequent analyses, the authors focus more on the GFP+ cells and find an enhanced migratory and invasive behaviour that goes along with increased mitochondrial activity. *In vivo*, GFP+ cells show a higher propensity for metastasis, and the proportion of QQ cells is higher in metastases compared to primary melanoma. Finally, they use their single cell data and bulk sequencing of Ki67-negative melanoma cells to identify markers for metastatic quiescent cells. Using this, they identified the surface protein GPNMB as strongly enriched in this population. GPNMB positive melanoma cells have similar *in vivo* properties to GFP+ cells, and the authors successfully used Glembatumumab vedotin (GV), a GPNMB-targeting cancer drug, to target metastasis development in the mouse melanoma models.

The study is highly interesting and was very carefully conducted and controlled, using two animal models and including publicly available datasets. It is particularly elegant that their findings come together in the application of a drug that is already used in clinical trials for several tumor entities including melanoma. Overall, the data are very convincing. The authors present a complex study encompassing a wealth of data. In my opinion, they address all major relevant research questions of this topic.

However, some questions arose that need to be addressed in order to better explain the methods or interpretation of the findings.

1) It would be useful to mention the presence of common melanoma driver and tumor suppressor genes of the melanoma cell lines in the Materials and Methods section.

As suggested by the reviewer, we added information regarding the common melanoma driver and tumor suppressor genes present in the melanoma PDXs to the Methods section. We would like to note that the two PDXs used in this study have been previously extensively characterized in Bossi et al. (2016), which is referenced in our manuscript.

2) Currently, the authors have only deposited their scRNA seq data under a private link (GSE251865), but they did not mention the deposition of the bulk RNA seq data. This should be amended, and a reviewer link for the datasets should be generated.

Thank you for bringing this to our attention. We apologize for the omission of the bulk RNA-seq data deposition. We uploaded the data to the repository and the reviewer link serves now for both the scRNA-seq and bulk RNA-seq datasets (GEO accession number: GSE288716).

3) Figure 1B/Supplementary Figure 2A (Appendix Figure S2A): Please explain shortly how the fitness (Hellinger distance) is interpreted. How would a perfect fit look like?

The goal of the fitness is to quantify the quality of the fitting of a model's simulation with respect to the experimental measurement. This evaluation is necessary to guide the swarm intelligence optimization process, and provides a final assessment of the quality of the model itself.

In this case, the data correspond to the fluorescence histograms. The Hellinger distance provides a (positive) real value, where a perfect fit of the two histograms (simulation, experimental data), bin by bin, is equal to zero. Due to the stochasticity of the simulations, though, a perfect match of the histograms is never really possible. The rule of thumb is: the lower the fitness, the better the model. Thus, if we look at Supplementary Figure 2A (Appendix Figure S2A), we can see that Model#1 - whose fitness never went below a value 11 across all tests - is worse than Model#2, which consistently gave a fitness slightly smaller than 5. Supplementary Figure 2B (Appendix Figure S2B) shows that the two histograms are indeed overlapping, except a few cells that can be easily explained by the stochasticity of the simulation or the experimental noise.

4) Figure 2D: The authors explain that "KI67 expression was used as a marker of active G1-S-G2M cell cycle phases, while BrdU incorporation was used as marker of DNA synthesis (24 hours pulse)." However, Figure 2D shows a lack of KI67 expression in G1 cells. Can the authors comment on the question of sensitivity (how much of the cell cycle phase is covered by these treatments)? How likely is it to miss a signal in a cycling cell?

The G1 phase in Figure 2D represents G0 (KI67-) and G1 phases. KI67 is degraded continuously in G1 and G0 phases so the level in individual cells is highly heterogeneous and depends on how long each cell has spent in G0. Given the dynamic nature of KI67 expression, the violin plot representing the data of Figure 2D (see below) depicts a wide and sensitive distribution of KI67 levels across the cell cycle phases, suggesting that we can accurately capture the range of KI67 levels. Figure 2E provides evidence that KI67 is detected in some cells within the G1 phase. Indeed, imposing the threshold on the median expression, even if it might be slightly underestimating the data, there is a percentage of KI67 positive cells falling in the G1 area that can definitely fit with the percentage of KI67+ and BrdU+ that we see at protein level in the GFP+ population in Figure 1C-D.

E

5) Text to Figure 2 and Supplementary Figure 3 (Appendix Figure S3): "Analyses of several published melanoma signatures showed increased expression of invasion-related pathways ("MITF targets", "Invasion" and "MSC" melanoma signatures) in GFP+ cells, and of proliferation- and melanocyte-associated pathways ("Mitotic" and "Pigmentation" signatures) in GFP- cells (Fig. S3A)". This is hard to see in the UMAP visualisation, as GFP+ and GFP- segregate minimally (as the authors state themselves). I would recommend to omit the UMAP visualisations and only refer to the violin plots.

As suggested by the reviewer, the UMAP visualization is not very informative and we delete it from the manuscript and retained only the violin plots.

6) Figure 2D Figure legend: "UMAP visualization of 14,353 MM13 melanoma cells analysed by scRNA-seq and integrated across 3 different melanoma primary tumors with 4 matched lymph node metastatic lesions...". It only becomes clear much later in the paper (Figure 7) where the primary tumors and lymph node metastases are coming from. Please refer to this section of the paper to avoid confusion.

We acknowledge the reviewer's concern regarding the clarity of the tumor origins. To address this, we have revised the second paragraph of the Results section to explicitly state the source of the primary tumors and lymph node metastases (highlighted in red in the text). We have also included a reference to Figure 7 to provide additional context and prevent reader confusion.

7) Supplementary Figure 4E (Appendix Figure S4E): " Moreover, analyses of a recently published melanoma signatures from patient biopsies (52) showed that our QQ/PQ cell states are enriched in the expression of melanoma states associated to acquired resistance to immunotherapy (Fig. S4E)". Please provide references describing the involvement of the signatures in acquired resistance to therapy. Also, which datasets were compared to the QQ data to calculate the indicated p values?

Pozniak et al. (ref. 52) demonstrated that a mesenchymal-like (MES) state in melanoma cells is significantly enriched in early on-treatment biopsies from non-responders to immune checkpoint blockade (ICB), indicating its association with acquired resistance. Analyzing the expression of their whole dataset with our QQ/PQ/QP/PP signatures, we highlighted a significant enrichment of the MES state in our QQ cells.

8) Figure 3C: "Conversely, State 3 was enriched in QP and PP cells". According to the legend, showing states 1,2, and 3, it seems to me that this statement is be more relevant for state 2. However, the colour legend is tiny, and it is not easy to differentiate between

green and blue (at least for me). Please check this statement and increase the size of the colour legend.

Shortly after this (page 5 of the Results section), the authors write: "...transiting from a quiescent (QQ) to an active proliferative state (PP and PQ), and back to a quiescent (PQ) state." It seems that "(PP and PQ)" contains a typo that should be corrected to "(PP and QP)".

We apologize for the mistakes. We have corrected the statement to accurately reflect the data, changing "State 3" to "State 2", as the reviewer pointed out. We also increased the size of the color legend. Similarly, we have corrected the typo on the same page, changing "PP and PQ" to "PP and QP".'

9) Fig. 4D; Fig. S5D: "Furthermore, comparison of the mitochondrial respiratory activities of freshly isolated MM13 and MM27 GFP+ and GFP- cells revealed increased mitochondrial respiration in GFP+ cells, with no change in mitochondrial mass". What is the data basis for the latter statement ("no change in mitochondrial mass")? Please comment.

We apologize for the missing data in the figure, which has been included in the revised version. Mitochondrial activity (Mitotracker orange) and mass (Mitotracker Deep-Red) were measured in MM13 (see below and new Fig.4D) and MM27 (see below and new Appendix Fig. S5D) GFP+ and GFP- cells, revealing no significant differences in the mitochondrial mass. Final mitochondrial activity was determined by calculating the ratio of the mean fluorescence intensity (MFI) for mitochondrial activity to the MFI for mitochondrial mass.

MM13

MM27

10) Fig. 5E-F; Fig S6D-E: "GFP+ populations showing improved migratory and invasive properties in hypoxic conditions in the first 8-16 hours of the experiments...". This statement refers to the 2D experiments in Figure 5E and Figure S6D. In the 3D experiments, improved invasive properties are seen until 2-3 hours (Figure 5F, Figure S6E). Please correct this statement. Also, the resolution of Figure 5E needs to be improved.

We thank the reviewer and we corrected the statement related to the 3D experiment "Analyses of cell speed in the invasion assay showed an increased average speed of GFP+ cells up to 2-3 hours in low oxygen conditions (Fig. 5G; Fig. S6F)." We also improved the resolution of Figure 5E.

11) In Figure 6H and Figure S7G, the authors describe "larger metastases within organs, as evidenced by the increased number of human CD298+ cells quantified upon sacrifice (day 100 post injection)". In their figure legend, they define the metastasis size by "average number of organs with at least 50% of melanoma cells per mouse". This is a lot of tumor load in the respective organs, and it is surprising that the mice survived this long without the need for prior sacrifice. Could the authors please comment on this and explain in better detail how the analysis of metastases was performed and how the numbers were calculated (in a three-dimensional organ context)?

Primary GFP+ and GFP- tumors were resected upon reaching 300 mm³. Following resection, mice were monitored for metastasis development. At 100 days post-injection, the experiment was terminated, and all mice were euthanized. Euthanasia was triggered when the first mouse in the group met the criteria outlined in our internal animal procedure protocol, specifically a lymph node volume of 800 mm³. Post-mortem analysis revealed significant inter-animal variability in lymph node volumes (Figure 6G). We were surprised to observe large metastatic lesions within the organs, despite the absence of overt signs of distress in the mice. These signs, as defined in our internal experimental protocol, included 10% weight loss, ataxia, reduced mobility, twitching, scratching, irregular breathing, diarrhea, swollen stomach, and neurological disorders. Following organ collection, metastatic tissues were dissociated into single cells, and an aliquot was stained for CD298PE for FACS analysis.

12) Figure 7F, G, Figure S8D: "Cell populations with primary QQ cells enriched at early stages of the trajectory, while metastatic QQ cells enriched at later points of pseudotime (Fig. 7F-G). Interestingly, the quiescent PQ cells showed an opposite trend

being less enriched in early stage in primary and more enriched at later stage in metastasis (Fig. S8D)." As the general pseudotime course, shown in the respective density plots in Figure 7G and S8D is more or less similar between QQ and PQ cells, I find it counterintuitive to call this an "opposite trend".

We thank the reviewer for the comment, we agree and we rewrite the sentence as followed:
"Cell populations with primary QQ cells enriched at early stages of the trajectory, while metastatic QQ cells enriched at later points of pseudotime (Fig. 7F-G). Interestingly, PQ cells showed a similar trend, albeit with less pronounced differences between early and late stages in primary tumors, and more pronounced differences in metastasis (Appendix Fig. S8D)."

13) Figure 8B: "GPNMB was highly expressed in QQ and PQ cells compared to other cell states (Fig. 8B)." A reference to Figure 3A would be useful here.

We added a reference to Figure 3A, as suggested by the reviewer.

14) Figure 8F, Figure S9D: "Indeed, the percentage of GPNMB+/KI67- cells rises to 91,3% in the lymph node metastases". Do these figures show one example of a primary tumor and a corresponding lymph node metastasis, or are the data derived from pooled samples?

These figures show one example of a primary tumor and a corresponding lymph node metastasis. We clarified it in the figure legend.

Typos:

- 1) Introduction, page 1: „, are though to play a crucial role..." Should be: "... are thought to play a crucial role..."
- 2) Legend to Supplementary Figure 1C (Appendix Figure S1C): "...show cell proliferation (gradual GFP lost) over time...) Should be(gradual GFP loss)..."
- 3) Figure S9D: Typo in the labeling of the GPNMB+ LN met cells: lower section should be Ki67-
- 4) Last page of the Discussion: "Our research strengthens this possibility by identifying GPNMB as a specific marker for targeting the pro-metatstatic and drug resistance phenotypes of quiescent melanoma cells." Should be: "pro-metastatic"

We thank the reviewer for highlighting the typos: we fixed them in the manuscript.

Referee #2:

In the context of cancer, quiescence is thought to concern a small sub-subpopulation of cells within the tumors that is capable of reverting to a growth state and leads to both relapse and drug resistance. In this manuscript, Lotti et al challenge this concept in cutaneous melanoma by showing that quiescence is accessible to most, if not all, tumor cells. They also provide a comprehensive analysis of quiescent melanoma cells and identify a novel (GPNMB) marker of this cell state that can be targeted to mitigate pro-metastatic ability and development of drug resistance. Specifically, Glembatumumab vedotin, a GPNMB inhibitor, induces the death of quiescent melanoma cells.

This article contains a large amount of data. It well-written and experiments have been carefully designed. The model system is extremely well-described and has been used successfully in previous publications by the same group. In my opinion, the results presented here may also be of interest for understanding the process of quiescence, metastatic spread and drug resistance in other cancer models.

Below are my comments to be considered by the authors.

Results:

Page 9:

The authors wrote "In melanoma, low oxygen (LO) conditions drive the ability of melanoma cells to switch between different states, to develop therapy resistance and to exhibit increased expression of markers associated with cancer stemness, particularly those related to slow-cycling properties". I would suggest the authors to add the reference "Cheli et al" (Oncogene 2012, PMID: 21996743) which is highly related to this section.

We appreciated the reviewer's insightful suggestion. The recommended reference is indeed highly relevant. We have added it (highlighted in red in the text) and expanded our discussion based on the reviewer's other comments.

Figure 8:

- Do oxygen concentrations influence GPNMB expression?
- Is forced expression of GPNMB in GPNMB-negative cells sufficient to render them more quiescent and affect their motility or MEKi sensitivity?

We thank the reviewer for their inquiry regarding the influence of oxygen concentrations on GPNMB expression. To explore this, we cultured MM13 and MM27 PDX cells under normoxic (21%) and hypoxic (3%) conditions for three days. Subsequently, we performed FACS analysis to quantify GPNMB surface expression. Our results demonstrate that hypoxic conditions induced an increase in GPNMB membrane expression in both PDXs, although the magnitude of this increase varied. We repeated the experiment twice and here we show a representative FACS analysis from one experiment and the plot of 2 independent experiments.

Furthermore, in collaboration with an experienced pathologist, we performed spatial analysis of GPNMB-positive cells within the tumor. This revealed an increased percentage of these cells in hypoxic areas, particularly those adjacent to necrotic regions (see response to Referee 3 below).

We acknowledge the reviewer's suggestion to investigate the effect of GPNMB overexpression in GPNMB-negative cells. However, extensive literature demonstrates that GPNMB overexpression, across various tumor types, promotes cancer cell migration, invasion, and metastasis, and contributes to drug resistance and the enhancement of stem-like properties (Chen et al., 2018; Rose AAN et al., 2017; Okita et al., 2018; Jian-Po Zhai et al., 2021; Maric et al., 2015; Rose et al., 2010; Okita et al., 2017; Fiorentini et al., 2014). Therefore, we anticipate that ectopic GPNMB expression in GPNMB-negative cells would likely induce a more aggressive, drug resistant, and quiescent, stem-like phenotype. Furthermore, this experiment presents significant technical challenges. Accurate quantification of ectopic GPNMB expression, particularly its cellular localization, would require a complex set-up. Given these technical hurdles and the well-established role of GPNMB in promoting these phenotypes, we believe that pursuing this challenging experiment would be beyond the current manuscript's goal.

Figure 8F: The authors show that GPNMB expression is associated with both quiescent and metastatic phenotype, with the percentage of GPNMB+/KI67- cells rising to 91,3% in the lymph node metastases. Is the percentage of GPNMB+/KI67- cells is also up-regulated in other metastatic sites (lung, liver...)? Can the authors comment on this?

We thank the reviewer for the question. To address this point, we analyzed GPNMB+/KI67- expression in lung metastases matched with the lymph node and primary tumor samples shown in Figure 8F. As shown below, our analysis revealed a higher percentage of GPNMB+/KI67- cells in lung metastases (87.4%) compared to the primary tumor (81.5%), reinforcing the association between GPNMB expression and both quiescent and metastatic phenotypes. While we acknowledge the relevance of this finding, we have opted not to include it in the manuscript, as our primary focus is on lymph node metastasis.

Figure 8O: While GPNMB negative cells have a significantly shorter latency to resection (fig. 8L), the authors did not observe the same effect when inhibiting GPNMB with the ADC? Can the authors comment on this point?

We believe the discrepancy in observed effects is attributable to the experimental design. Figure 8L represents an *in vivo* experiment using mice injected with sorted GPNMB+ and

GPNMB- populations, whereas Figure 8O utilized bulk cell injections. This difference likely resulted in a dampening of the observed latency differences.

Discussion:

Page 15: The authors show that primary-tumor QQ cells displayed enriched pathways related to iron ion homeostasis. Would a ferroptotic drug be of interest in the adjuvant setting? This point could be discussed in the discussion section.

We thank the reviewer for bringing up this interesting point. We have expanded the discussion section to address it.

“While hypoxia conditions are known to play a role in pushing cells towards quiescence (Emami Nejad et al., 2021; Fattore et al., 2020; Muz et al., 2015; O’Connell et al., 2013; Roesch et al., 2010), the iron ion homeostasis has been linked to cancer stemness maintenance (Guo et al., 2021; Katsura et al., 2019; W. Wang et al., 2022). Furthermore, both oxidative stress and iron metabolism are linked to ferroptotic process, suggesting that quiescent melanoma cells could be sensitive to ferroptosis inducing agents (Perego et al., 2018). Iron overload, antioxidant inhibitors or combination of ferroptosis inducers with traditional therapies can disrupt the cellular antioxidant defense system, providing a new therapeutic approach for targeting metastatic quiescent cells to prevent relapse, especially in the still under explored adjuvant setting (Ta et al., 2023; Zhou et al., 2024).”

Page 16: The authors suggest that the definition of the quiescent state and of the pathways activated/deactivated in these cells may vary between cell lines or patient-derived cells, and this definition still requires further characterization. Related to this sentence, I would suggest the authors to discuss the manuscript from "Cheli et al. (Oncogene 2011 PMID: 21278797)". Cheli et al showed that (i) slow-cycling CFSE retaining cells have an increased capability of migrating in vitro, (ii) displayed an increased p27 expression, which is required for exacerbation of the tumorigenic properties of melanoma cells and (iii) increased expression of SPARC, but (iv) displayed low MITF expression.

Could it be that slow-cycling CFSE retaining cells (Cheli et al. PMID: 21278797)" and GFP+ cells (Lotti et al herein) share molecular features (enhanced motile ability) and markers (p27, SPARC) but represent different stages in the transition towards the quiescent state with also different markers (MITF, GPNMB)?

We appreciate the reviewer's suggestion to discuss our findings in the context of Cheli et al. (Oncogene 2011, PMID: 21278797). As the reviewer highlighted, Cheli et al. demonstrated that slow-cycling CFSE-retaining cells exhibit enhanced migratory capacity, increased p27 and SPARC expression, and decreased MITF expression. We observed similar characteristics in our quiescent cell populations: increased in vitro migration (Fig. 4A, Appendix Fig. S5A), elevated p27 (Appendix Fig. S4C) and SPARC expression. However, as shown below, direct comparison of the Cheli et al. gene signature with our QQ and PQ cell signatures revealed limited overlap. Specifically, only three genes were shared with our PQ signature, and no genes were shared with our QQ signature. This discrepancy may be attributed to the differing methodologies employed: Cheli et al. utilized a TaqMan stem cell array, while we performed single-cell RNA sequencing.

Therefore, while our quiescent cells and the slow-cycling CFSE-retaining cells described by Cheli et al. share molecular features and markers, they may represent distinct cell states along the trajectory towards quiescence. This warrants further investigation to fully elucidate the nuances of quiescence in melanoma.

Minor:

Fig7C (right panel): I would suggest swapping KI67 expression in the primary tumor and metastases in the graph to follow the same order as in the other panels.

We thank the reviewer for the suggestion. We have reordered the graph to match the other panels, displaying primary tumor data before metastasis data (see below and corrected Fig. 7C).

Figure 8: Could the authors please indicate on the figure panels and also in the figure legend which panels correspond to the QQ and PQ cells.

We apologize for the ambiguity in Figure 8. Figure 8G shows GPNMB expression in primary and metastatic MM13 QQ cell populations while all the other panels show bulk cell analysis. Supplemental Figure 8 (Appendix Figure S8) details MM13 PQ cell analysis. We have updated the figure panels accordingly.

Figure 8N: Can the authors describe in the "NT" caption (I assume these are non treated cells).

The reviewer is correct, NT stands for non-treated cells. We apologize for the missing caption information. This is corrected in the revised version.

Referee #3:

This manuscript uses a tagging approach to identify quiescent cells in xenograft tumours and functionally assess their metastatic ability. Essentially, they use a dox repressible system to turn off H2b-GFP expression and define quiescent cells as the GFP hi cells after some time. There is a lot of important detail missing from these experiments that makes it difficult to understand some parts of this manuscript. The potential shortcoming of this system is that it requires doxycycline to repress marker gene expression.

The models are then grown as xenografts to enhance the biological relevance of the models, but this may also introduce unwanted problems. We do not know if repression of the entire tumour can be maintained in vivo.

The authors identify apparently hypoxic regions within the tumour which is a common feature of xenografted models and regions of poor blood vessel infiltration. Thus, some of the GFP+ cells may still represent proliferating cells where derepression has occurred. Also as recognised by authors, the tumour is in dynamic change with quiescent cells re-entering the cell cycle and cycling cells exiting into quiescence.

We appreciate the reviewer's concerns regarding the H2B-GFP tagging approach used in our study. We acknowledge that, as with any experimental model, this system has inherent limitations. While we recognize the potential for derepression of GFP expression in vivo, particularly in hypoxic regions, we have taken steps to mitigate this issue. The consistent percentage range of GFP+ cells observed across multiple chasing experiments provides evidence for the robustness of our in vivo system in maintaining GFP repression. Furthermore, the successful isolation of two functionally distinct populations (GFP+ and GFP-) from two different PDX models, as demonstrated in Figures 1 and Supplemental 1 (Appendix Figure S1), supports the validity of our approach. We acknowledge that the GFP+ population may not represent a pure quiescent state, as indicated in Figure 2 and supported by our mathematical model. However, this system remains a valuable tool for isolating and characterizing label-retaining and non-label-retaining cell populations, allowing us to further characterize and dissect them. The presence of varying KI67 expression within both GFP+ and GFP- populations highlights the dynamic nature of tumor cell states and the continuous transition between quiescence and proliferation. This observation is central to our study and provides valuable insights into the complexity of the quiescent melanoma population.

We understand the reviewer's concerns regarding the use of xenograft models and the potential for artifacts. However, these models offer a crucial in vivo context for studying metastatic ability.

After figure 1 where they show this dynamic effect, they go on to use better markers of quiescence such as BrdU incorporation and Ki67 expression but persist with the broad classification of GFP+ and GFP- representing quiescent and proliferating cells. I think that this leads to a lot of unnecessary confusion. The subsetting of cells in QQ, PP, QP, PQ for example does not really give any new insights and I think is flawed.

We understand the reviewer's concern regarding the potential for confusion stemming from the broad classification of GFP+ and GFP- cells.

While we acknowledge that BrdU incorporation and Ki67 expression are established markers of quiescence, our data strongly suggest that the pro-metastatic phenotype of GFP+ cells is directly linked to the enrichment of quiescent cells within this population, as evidenced by the predominant negativity of Ki67 and BrdU. However, Ki67's nuclear localization precludes its use for purifying viable quiescent cells for functional assays. To overcome this limitation, we

performed transcriptional profiling of Ki67+ and Ki67- cells within both GFP+ and GFP- populations. This analysis aimed to identify surrogate surface markers that would enable the isolation and functional characterization of these distinct cell populations.

The subsequent subsetting of cells into QQ, PP, QP, and PQ categories was a direct result of this transcriptional analysis, and while we understand the reviewer's concern that this may not provide "new insights", it was a necessary step to find surface markers that allowed us to functionally test our hypothesis. We believe that this approach, while complex, provides a valuable framework for understanding the heterogeneity and functional diversity of quiescent melanoma cells.

The expression analysis is performed at a single time point (I think that is the case, the details of this were lacking) so are a snapshot of cells at that time. Unless QP and PQ subset cells are caught just as they are transitioning from quiescence to proliferation or the reverse, it is difficult to understand how they can have significantly different transcriptional programs.

We acknowledge the reviewer's point that the expression analysis, performed at a single time point, represents a snapshot of the cellular transcriptome.

Indeed, this snapshot captures a moment of dynamic transition, where varying KI67 levels (high in QP and low in PQ) are actively shaping the transcriptional program. We hypothesize that the observed transcriptional differences between QP and PQ cells reflect this transitional state. It is important to note that, despite these dynamic changes, PQ and QQ cells exhibit significant overlap in stress, cell death, and therapy resistance signatures, which distinguishes them from the proliferative QP and PP populations. This shared transcriptional profile suggests a common underlying biology for these quiescent-associated states.

In Figure 3A, the UMAPS show again that GFP status is a very poor indicator of quiescence or proliferation. The Ki67- GFP-+ population overlap completely, as do the Ki67+ populations in Fig 3A. I think the data in Fig 3B is confounded by the GFP+ Ki67+ being derepression of GFP expression in proliferating cells, and cells that have lost GFP expression possibly and exited cell cycle.

The observed overlap of GFP+ and GFP- populations within both KI67+ and KI67- cells reflects the dynamic transitions occurring between cell states. This observation is central to our study and highlights the complexity of defining quiescence in melanoma.

While we recognize the potential for GFP derepression in proliferating cells, as suggested by the reviewer, our data indicate that the observed Ki67+ GFP+ population is not solely attributable to this phenomenon. The presence of Ki67- GFP- cells, which are unlikely to be experiencing GFP derepression, further supports the notion of dynamic cell state transitions. Moreover, the observation that cells can lose GFP expression and exit the cell cycle, as the reviewer suggests, aligns with our model of dynamic quiescence. This process contributes to the observed heterogeneity and overlap between GFP and Ki67 status.

We believe that the data presented in Figure 3B, while complex, accurately reflects the dynamic interplay between proliferation and quiescence in our model. The observed patterns are not simply artifacts of GFP derepression or loss, but rather reflect the inherent plasticity of melanoma cells.

The pathways identified in the PP and QP cells are almost identical, and overlap better than the QQ and PQ gene sets. If these different populations represented some particular spatial region of the tumour e.g. in the vicinity of hypoxic regions this might

provide support for these dynamic populations representing functionally different forms.

We appreciate the reviewer's suggestion regarding the potential spatial distribution of the identified cell populations and its functional implications, a point that echoes a similar comment from Reviewer 1 (see above).

Direct spatial visualization of the four cell populations (QQ, PP, QP, PQ) is technically challenging because these populations represent transient states, and we lack specific markers for their precise identification, except for the QQ population. We acknowledge, however, the considerable potential value of such an experiment in understanding their spatial organization. To indirectly address the reviewer's point, we performed a spatial analysis of our combined quiescent population (QQ and PQ), defined by GPNMB expression, in relation to hypoxic/necrotic areas, identified by an experienced pathologist, within primary MM13 tumors. This analysis, that is now detailed in Appendix Fig. S9E, provides insights into the spatial relationship between quiescence and hypoxia within the tumor microenvironment. Indeed, the number of GPNMB-positive cells showed a 2,3-fold increase in the peri-hypoxic regions, compared to the normoxic counterparts (6.5% in normoxic areas and 14.8% in peri-hypoxic regions; $n=2$, $p=0,039$ unpaired t -test), defining a preferential spatial localization of GPNMB+ cells within the primary tumor. While this approach does not directly address the spatial distribution of all four cell populations, it provides valuable information regarding the spatial context of our quiescent cells and their relationship to hypoxic regions, which may contribute to their functional differences.

The major conclusion from this study is that less proliferative/quiescent cells have an enhance metastatic potential. This is very similar to the MITF low MITF high effect. Is this really any different?

We understand the reviewer's concern that our observation of enhanced metastatic potential in less proliferative/quiescent cells may parallel the MITF low/high paradigm.

MITF is a crucial regulator of melanoma survival, differentiation, proliferation, invasion and senescence (Goding & Arnheiter, 2019). It controls various cellular processes, including ABCB5 transporter expression (Louphrasitthiphol, Chauhan, & Goding, 2020), pigmentation, cell-cycle progression, and metabolism (Carreira et al., 2005, 2006; Du et al., 2004; Loercher et al., 2005; McGill et al., 2006; Louphrasitthiphol et al., 2019; Haq et al., 2013; Vazquez et al., 2013; Vivas-Garcia et al., 2020). MITF expression is suppressed by microenvironmental cues like hypoxia, nutrient limitation, and inflammation (Cheli et al., 2012; Louphrasitthiphol et al., 2019; Widmer et al., 2013; Falletta et al., 2017; Ferguson et

al., 2017; Riesenber g et al., 2015). MITF-high cells are typically proliferative or differentiated, while MITF-low cells exhibit increased invasiveness, drug resistance, and higher tumor-forming capacity (McGill et al., 2006; Widlund et al., 2002; Carreira et al., 2005, 2006; Loercher et al., 2005; Konieczkowski et al., 2014; Muller et al., 2014; Cheli et al., 2011). However, in the same cells MITF may repress or activate different sets of genes (Goding & Arnheiter, 2019).

While MITF's role in quiescence is still poorly understood, our GPNMB-positive quiescent melanoma cells showed high MITF, Tyr, and ABCB5 expression (Appendix Fig. S9I), suggesting a differentiated state. This highlights a paradox: GPNMB, a stem cell marker usually linked to undifferentiated cells, is also associated with differentiated, MITF-high melanoma cells, which should be proliferative. These quiescent cells, regardless of MITF levels, may possess aggressive traits like increased invasiveness, drug resistance, and tumor-initiating potential. This mirrors ABCB5, which also marks slow-cycling, differentiated tumor-initiating cells. We discussed these data in the last session of the Discussion.

Therefore, while there may be some overlap in the observed phenotypes, we identified a quiescence cell state that, even though expressing melanocytic differentiation markers (MITF; Appendix Fig. S9I), is characterized by drug resistance and invasive/EMT traits (Appendix Fig. S9G-H), distinct from the MITF high state described in the previous studies.

We believe that our model is not MITF related and offers a complementary perspective on the role of quiescence in melanoma,

Finally, the identification of GPNMB is interesting although the authors have not clearly stated their purpose here. Also, in Fig 8H, there is a subset of Ki67+ mets with very high GPNMB suggesting it is not the strongest candidate for a marker of quiescent cells. The other data generated show statistical significance but there is little quantitative difference observed.

While the study's primary focus in identifying GPNMB is not explicitly stated, its discovery offers a valuable tool for isolating and studying quiescent, invasive melanoma cells. Figure 8B-G show a statistically significant inverse correlation between GPNMB expression and Ki67 in the quiescent population, with GPNMB statistically different between groups.

Figure 8H shows that GPNMB is significantly enriched in Ki67- metastases compared to Ki67+ metastases and primary tumors, highlighting its clinical relevance. Therefore, GPNMB emerges as a promising marker for isolating quiescent melanoma cells and a potential therapeutic target for metastatic progression.

The strongest data is the GPNMB directed ADC which clearly blocks metastasis and is strong evidence that GPNMB expressing cells are a metastatic population. Indeed, this is a very clear demonstration that quiescent cells are indeed more metastatic than their proliferating counterparts. If this was made more clearly the conclusion of this study it would be suitable for publication in this journal.

We thank the reviewer for the appreciation and, as suggested, we clarified and emphasized this data in the Results and Discussion sessions of the manuscript.

As it is at present, the authors present a very jumbled story that is very difficult to follow, and makes unnecessary claims not strongly backed by evidence.

Specific points:

Figure 1C-E; Figure 3; at what time point were these cells analysed? 3 weeks after Dox addition? This information must be in the figure legend. There is no indication what

time after dox induced repression the scRNAseq data in figure 2 was generated? Are the cells isolated from PDX generated under the same conditions and time of exposure to dox as the scRNAseq experiments? If so, the GFP+/- subsetting is a flawed approach as indicated by the scRNAseq analysis.

We apologize for the missing information that has now been added in figure legend.

Figure 1C-E and the scRNAseq data were generated from cells isolated from two different PDXs after doxycycline treatment for 3 weeks (primary tumors) or 6 weeks (matched lymph node metastases). The chosen treatment periods resulted in two distinct, sortable cell populations (GFP+ and GFP-) that displayed biologically different phenotypic profiles.

Figure 2 refers to the analysis of 3 different primary melanoma tumors with 4 matched lymph node metastatic lesions, both groups treated as described above. In Figure 3, we used the same cells isolated from PDX generated under the same conditions and exposure times to doxycycline as the scRNAseq experiments described in Figure 2. We specified this in figure legend. Acknowledging the inherent limitations of the H2B-GFP model for isolating pure quiescent cells, we employed KI67 to define a quiescent signature, which facilitated the identification of GPNMB as a reliable marker for direct isolation.

Figure 4B; The reduced sensitivity is likely because the the GFP+ cells are not proliferating and thus likely to be less sensitive. Proliferating cells are more sensitive to MEKi.

Figure 4C: the reduced MEKi sensitivity in the GFP+ cells can be explained by their lack of proliferation, as proliferating cells are generally more susceptible to MEKi. Nevertheless, since the viability data are normalized to untreated samples, the observed resistance, combined with the enhanced migratory and invasive properties of the GFP+ cells, indicates a unique and feature of this population.

Figure 4D; The mitotracker data indicates that there is more mitochondrial mass in the GFP+ cells. As the data in Fig 4D is from a double staining of MitoTr Orange that measures mitochondrial activity and MitoTr far red measuring mito mass, my guess is that the histogram is the far red and the bar graph is the ratio of orange and red signals. How this is not defined clearly in the legend. Why not show this data as a dot plot of the GFP- and GFP+ on the same graph of Red v orange fluorescence intensity. I cannot understand the data as it is presented currently.

We appreciate the reviewer's feedback, which, consistent with Reviewer 1's observations, led us to include the essential mitochondrial mass data, thereby providing a more comprehensive understanding of our experimental conclusions. Mitochondrial activity (Mitotracker orange) and mass (Mitotracker Deep-Red) were measured in MM13 (see below and new Fig.4D) and MM27 (see below and new Appendix Fig. S5D) GFP+ and GFP- cells, revealing no significant differences between the mitochondrial masses. Final mitochondrial activity was determined by calculating the ratio of the mean fluorescence intensity (MFI) for mitochondrial activity to the MFI for mitochondrial mass.

MM13

MM27

Figure 4E; If the MitoTr red (PE) is showing more mitochondria then the increased O2 consumption is an expected result. The difference in mitochondrial mass is surprising.

We have clarified that there is no difference in mitochondrial mass, addressing the reviewer's concern regarding the MitoTr red (PE) data and O2 consumption.

Figure 5G; MITF hi, G1 phase cells are known to be faster migrating.

Figure 5G demonstrates that GFP+ cells, enriched for quiescent cells, exhibit higher migration speeds than GFP- (proliferating) cells under both controlled (COC) and standard (SOC) oxygen conditions. This enhanced migration, consistent with the in vivo observations we described above (GPNMB+ cells in tumor tissue sections), suggests that quiescent cells are highly adaptable to low-oxygen, acquiring invasive traits. We also reiterate that MITF in these cells is associated with differentiation and stemness, not with quiescence, which is distinct from previous findings, as discussed above.

Figure 7A; Problem with the %GFP+ cells in lymph nodes in the much higher background fluorescence in that sample. This could significantly affect the % GFP+. A control of lymph node from non-tumour bearing mice would provide a better cut-off.

We sincerely apologize for the error in Figure 7A. We mistakenly presented an incorrect dot-plot for the lymph node metastasis. We have now replaced it with the accurate imageshows that there is no significant difference in background fluorescence compared to the primary tumor, as lymph node metastases are composed almost entirely of the primary cancer cells, with minimal to no contribution from lymph node tissue.

Figure 7B; Is it a surprise that primary and mets have different expression profiles? I would expect to see difference in both the KI67- and KI67+ population between primary and mets as mets are in a very different environment and therefore must change their transcriptome to allow them to survive in the new environment.

As the reviewer highlights, it's not a surprise that primary and metastatic site have different environments so different expression profiles. The interesting point, which might not be so expected, is that the metastatic environment promotes an enrichment of the QQ (GFP+KI67-) population, shown in Fig.7B.

Figure 7c; A big difference here is the GFP+ primary and GFP- met have lost their cluster 0 from figure 2A, but cluster 0 is very prominent in the other population in each case.

To provide a clearer understanding of the cluster differences in Figure 7C, we have generated clusterization data for GFP+ and GFP- cells from primary tumors and metastases. Stack plots are included to show the relative distribution of these populations across clusters, revealing transcriptional differences and the presence of cluster 0. To further illustrate the distinct nature of these populations, we've also subclustered the QQ (GFP+KI67-) cells in Figure 7D, which highlights the significant differences.

Page 12, 2nd paragraph; I don't understand your conclusion? How is this promoting dormancy? This isn't a lot of difference between primary and mets in terms of Ki67-cells.

Our conclusion that the metastatic microenvironment is promoting dormancy is supported by the enrichment of the GFP⁺ population in the metastatic site (Figure 7A). Furthermore, we observed a statistically significant decrease in KI67 expression in QQ metastatic cells compared to QQ primary cells (Figure 7C), indicating a reduction in proliferation and thus, an increase in dormancy.

Figure 5E legend; is this 3 experiments or 3 cells?

We fix the figure legend clarifying that 3 is the number of the experiments we performed.

Dear Dr. Lanfrancone,

Thank you for the submission of your revised manuscript to our editorial offices. I have now received the reports from the three referees that were asked to re-evaluate the study, you will find below. As you will see, the referees now support the publication of your manuscript in EMBO reports. Referees #1 and #3 have remaining minor concerns and suggestions to improve the manuscript, I ask you to address in a final revised manuscript. Please provide a final p-b-p-response to the remaining issues and the editorial requests below.

I have these editorial requests:

- Please provide a more comprehensive final title without colon and with not more than 100 characters (including spaces).
- Please list the authors First Name - Last Name on the title page, as in the submission system.
- Please remove the running title and the significance statement from the manuscript.
- Please provide the abstract written in present tense throughout.
- Please check again that the number "n" for how many independent experiments were performed, their nature (biological versus technical replicates), the bars and error bars (e.g. SEM, SD) and the test used to calculate p-values is indicated in the respective figure legends. Please also check that all the p-values are explained in the legend, and that these fit to those shown in the figure. Please provide statistical testing where applicable. Please avoid the phrase 'independent experiment' but clearly state if these were biological or technical replicates. Please also indicate (e.g. with n.s.) if testing was performed, but the differences are not significant. In case n=2, please show the data as separate datapoints without error bars and statistics. See also:
<http://www.embopress.org/page/journal/14693178/authorguide#statisticalanalysis>
- If n<5, please show single datapoints for diagrams. Moreover:
 - Please note that the exact p values are not provided in the legends of figures 1C-E; 3A, 4A-E; 5A-C, F, G; 6A, B, C, D, F, G, H; 7A, C; 8D, E, G, I, J, K, L, M, N, P, Q.
 - Please indicate the statistical test used for data analysis in the legends of figures 3B, 7H.
 - Please note that the box plots need to be defined in terms of minima, maxima, centre, bounds of box and whiskers, and percentile in the legend of figure 1B.
 - Please note that the box plots need to be defined in terms of minima, maxima, centre, bounds of box and whiskers in the legend of figure 8H.
 - Please note that information related to n is missing in the legends of figures 3A, 6C, D, E, F, G, H; 7A.
- Please add scale bars of similar style and thickness to microscopic images, using clearly visible black or white bars (depending on the background). Please place these in the lower right corner of the images themselves. Please do not write on or near the bars in the image but define the size in the respective figure legend. Presently, some scale bars (e.g. in panel 8I) are hard to see and need to be thicker. Please check.
- Please provide the primer information (Quantitative RT-PCR) only in the Reagents & Tools Table and remove the primer information from the Methods section.
- Appendix Tables S1 and S3 are Datasets. Please remove these from the Appendix and upload them as separate dataset files (named Dataset EV1 and Dataset EV2), with a legend on the first TAB of the Excel files. Please update their callouts, rename the remaining three Appendix Tables and also update their callouts.
- Please provide a 'clean' final Appendix file without track-changes or colored fonts.
- Please remove now the referee token from The Data Availability section (DAS) and make sure that all datasets are public latest on the date of online publication of the study.
- Per journal policy, we do not allow 'data not shown', which is stated in the manuscript (e.g. page 18). All data referred to in the paper should be displayed in the main or Expanded View figures, or an Appendix. Thus, please add these data (or change the text accordingly if these data are not central to the study). See:
<https://www.embopress.org/page/journal/14693178/authorguide#unpublisheddata>
- Please use our reference format (et al needs to be used after 10 author names; DOIs should only be used for preprints and datasets that have not been published yet):
<http://www.embopress.org/page/journal/14693178/authorguide#referencesformat>

- Please remove the section 'CONTACT FOR REAGENT AND RESOURCE SHARING' from the manuscript text file.
- During our figure integrity check, we noted a reuse of the four microscopic panels in Fig. 4B in Fig. 5B. Please check. If this is intentional, please clearly state this reuse in the respective figure legends.

In addition, I would need from you uploaded separately:

Best,

Referee #1:

In their revised version of the manuscript, the authors have sufficiently addressed the reviewer's questions. I would merely advise to include a small section regarding the interpretation of the Hellinger distance into the manuscript itself (it is currently only addressed more specifically in the letter to the editor).

Referee #2:

The authors answered satisfactorily to my concerns and they provided some clarifications on the points highlighted. I have no further comments.

Referee #3:

The authors have responded to my questions and comments in detail. I believe the changes made have improved the manuscript and its clarity. I am still not fully convinced by their arguments against the derepression, but on the whole I think that the manuscript is a useful contribution to the literature. There are two points that I think need to be addressed before the manuscript can be submitted.

Figure 7c, it is not clear what the UMAP plots are showing? If this is showing KI67 as claimed there must be either an intensity coding missing or something? The associated violin plots are also uninformative. The KI67 staining intensity of the whole tumours? I am not sure what this adds.

Figure 7E, How is the % cluster calculated? It appears to be on the % of total cells in each cluster but as there were less primary cells the % is not a real reflection of the % of total counts.

Point by point response

Editorial requests:

- **Please provide a more comprehensive final title without colon and with not more than 100 characters (including spaces).**

GPNMB drives metastasis formation, challenging the quiescent paradigm in melanoma

- **Please list the authors First Name - Last Name on the title page, as in the submission system.**

It has been done

- **Please remove the running title and the significance statement from the manuscript.**

It has been done

- **Please provide the abstract written in present tense throughout.**

This is the new version of the abstract included in the manuscript:

Melanoma exhibits high intratumoral heterogeneity, characterized by a diverse population of cells undergoing dynamic transitions between cellular states. These adaptive changes enable melanoma cells to survive in the harsh tumor microenvironment, acquire drug resistance, and metastasize. One such state, quiescence, has been linked to both relapse and drug resistance, but its underlying biology and molecular mechanisms remain poorly understood. Our study challenges the conventional understanding of melanoma quiescence. Contrary to the notion of a rare, unique subpopulation, we demonstrate that quiescence is a highly dynamic state accessible to most, if not all, melanoma cells. This state is exquisitely sensitive to microenvironmental cues. We identify GPNMB as a marker of quiescence, expressed in both primary and metastatic tumors. GPNMB-positive cells exhibit a prometastatic phenotype and are enriched in metastatic sites, suggesting a potential role for quiescence in tumor dissemination. Our findings position GPNMB as a valuable marker for isolating quiescent melanoma cells and as a potential therapeutic target to tackle metastasis.

- **Please check again that the number "n" for how many independent experiments were performed, their nature (biological versus technical replicates), the bars and error bars (e.g. SEM, SD) and the test used to calculate p-values is indicated in the respective figure legends. Please also check that all the p-values are explained in the legend, and that these fit to those shown in the figure. Please provide statistical testing where applicable. Please avoid the phrase 'independent experiment' but clearly state if these were biological or technical replicates. Please also indicate (e.g. with n.s.) if testing was performed, but the differences are not significant. In case n=2, please show the data as separate datapoints without error bars and statistics. See also:**

<http://www.embopress.org/page/journal/14693178/authorguide#statisticalanalysis>

Moreover:

- **Please note that the exact p values are not provided in the legends of figures 1C-E; 3A, 4A-E; 5A-C, F, G; 6A, B, C, D, F, G, H; 7A, C; 8D, E, G, I, J, K, L, M, N, P, Q.**

- **Please indicate the statistical test used for data analysis in the legends of figures 3B, 7H.**

- **Please note that the box plots need to be defined in terms of minima, maxima, centre, bounds of box and whiskers, and percentile in the legend of figure 1B.**

- Please note that the box plots need to be defined in terms of minima, maxima, centre, bounds of box and whiskers in the legend of figure 8H.
- Please note that information related to n is missing in the legends of figures 3A, 6C, D, E, F, G, H; 7A.

All figure legends have been corrected, inserting the requested information.

Upon re-examination of the precise p-value measurements in Figure 5B, we identified that two measurements (one for each group) had not been copied into the Original Source Data table that we submitted. These have now been inserted, and a new table generated; the p-value presented in the figure remains unchanged.

- Please add scale bars of similar style and thickness to microscopic images, using clearly visible black or white bars (depending on the background). Please place these in the lower right corner of the images themselves. Please do not write on or near the bars in the image but define the size in the respective figure legend. Presently, some scale bars (e.g. in panel 8I) are hard to see and need to be thicker. Please check.

It has been done

- Please provide the primer information (Quantitative RT-PCR) only in the Reagents & Tools Table and remove the primer information from the Methods section.

It has been done

- Appendix Tables S1 and S3 are Datasets. Please remove these from the Appendix and upload them as separate dataset files (named Dataset EV1 and Dataset EV2), with a legend on the first TAB of the Excel files. Please update their callouts, rename the remaining three Appendix Tables and also update their callouts.

It has been done and the text changed accordingly.

- Please provide a 'clean' final Appendix file without track-changes or colored fonts.

It has been done

- Please remove now the referee token from The Data Availability section (DAS) and make sure that all datasets are public latest on the date of online publication of the study.

It has been done

- Per journal policy, we do not allow 'data not shown', which is stated in the manuscript (e.g. page 18). All data referred to in the paper should be displayed in the main or Expanded View figures, or an Appendix. Thus, please add these data (or change the text accordingly if these data are not central to the study). See:

<https://www.embopress.org/page/journal/14693178/authorguide#unpublisheddata>

As these data are not considered central to the manuscript, they have been removed, necessitating the deletion of the referring sentence in the Discussion.

- Please use our reference format (et al needs to be used after 10 author names; DOIs should only be used for preprints and datasets that have not been published yet):

<https://www.embopress.org/page/journal/14693178/authorguide#referencesformat>

It has been done

- Please remove the section 'CONTACT FOR REAGENT AND RESOURCE SHARING' from the manuscript text file.

It has been done

- During our figure integrity check, we noted a reuse of the four microscopic panels in Fig. 4B in Fig. 5B. Please check. If this is intentional, please clearly state this reuse in the respective figure legends.

We appreciate you bringing the duplicated microscopic panels between Figure 4B and Figure 5B to our attention. We confirm this oversight and have now replaced the four panels in Figure 4B with new, distinct images derived from the same experiment.

In addition, I would need from you uploaded separately:

- a short, two-sentence summary of the manuscript (not more than 35 words).

This study reveals that quiescence in melanoma is not a rare, static state but a dynamic and reversible condition accessible to most tumor cells, driven by microenvironmental cues. GPNMB marks this state and associates with prometastatic potential, making it a therapeutic target.

- two to four short (!) bullet points highlighting the key findings of your study (two lines each).

- Quiescence in melanoma is a dynamic state, not a rare subpopulation, making most cells vulnerable.
- GPNMB is identified as a novel marker of quiescent melanoma cells in both primary and metastatic tumors.
- GPNMB-positive cells exhibit a prometastatic phenotype and are enriched in metastatic sites.
- Targeting GPNMB with an antibody-drug conjugate (ADC) effectively reduces metastasis, highlighting a promising therapeutic strategy.

- a schematic summary figure as separate file that provides a sketch of the major findings (not a data image) in jpeg or tiff format (with the exact width of 550 pixels and a height of not more than 400 pixels) that can be used as a visual synopsis on our website.

The requested schematic summary figure has been prepared and is submitted as a separate file.

Referee #1:

In their revised version of the manuscript, the authors have sufficiently addressed the reviewer's questions. I would merely advise to include a small section regarding the interpretation of the Hellinger distance into the manuscript itself (it is currently only addressed more specifically in the letter to the editor).

We include a small section regarding the interpretation of the Hellinger distance into the manuscript (Pag. 22).

Referee #2:

The authors answered satisfactorily to my concerns and they provided some clarifications on the points highlighted. I have no further comments.

Referee #3:

The authors have responded to my questions and comments in detail. I believe the changes made have improved the manuscript and its clarity. I am still not fully convinced by their arguments against the derepression, but on the whole I think that the manuscript is a useful contribution to the literature. There are two points that I think need to be addressed before the manuscript can be submitted.

Figure 7c, it is not clear what the UMAP plots are showing? If this is showing KI67 as claimed there must be either an intensity coding missing or something? The associated violin plots are also uninformative. The KI67 staining intensity of the whole tumours? I am not sure what this adds.

Figure 7C shows UMAP spatial distribution of GFP positive and negative cells not on the whole samples, as in Figure 2B, but on primary tumors and matched metastases presented on separate panels. This separation allows us to highlight the differences among the two populations and the violin plot shows that the GFP+ metastatic population is enriched in KI67 negative cells suggesting that the metastatic niche is promoting the quiescent state. We fixed the figure legend to clarify the data images.

Figure 7E, How is the % cluster calculated? It appears to be on the % of total cells in each cluster but as there were less primary cells the % is not a real reflection of the % of total counts.

In the stack plot presented in Figure 7E, percentages are calculated as the proportion of QQ primary or QQ metastatic cells within each cluster, relative to the total number of cells in that specific cluster. We have added this clarification in figure legend.

Dr. Luisa Lanfrancone
European Institute of Oncology
Department of Experimental Oncology
via Adamello 16
Milano, MI 20139
Italy

Dear Dr. Lanfrancone,

I am very pleased to accept your manuscript for publication in the next available issue of EMBO reports. Thank you for your contribution to our journal.

Yours sincerely,
